# Neural Networks as Kernel Learners: The Silent Alignment Effect

**Alexander Atanasov**\* , **Blake Bordelon**\* **& Cengiz Pehlevan**
Harvard University
Cambridge, MA 02138, USA
`{atanasov,blake_bordelon,cpehlevan}@g.harvard.edu`

## Abstract

Neural networks in the lazy training regime converge to kernel machines. Can neural networks in the rich feature learning regime learn a kernel machine with a data-dependent kernel? We demonstrate that this can indeed happen due to a phenomenon we term *silent alignment*, which requires that the tangent kernel of a network evolves in eigenstructure while small and before the loss appreciably decreases, and grows only in overall scale afterwards. We empirically show that such an effect takes place in homogenous neural networks with small initialization and whitened data. We provide an analytical treatment of this effect in the fully connected linear network case. In general, we find that the kernel develops a low-rank contribution in the early phase of training, and then evolves in overall scale, yielding a function equivalent to a kernel regression solution with the final network's tangent kernel. The early spectral learning of the kernel depends on the depth. We also demonstrate that non-whitened data can weaken the silent alignment effect.

## 1 Introduction

Despite the numerous empirical successes of deep learning, much of the underlying theory remains poorly understood. One promising direction forward to an interpretable account of deep learning is in the study of the relationship between deep neural networks and kernel machines. Several studies in recent years have shown that gradient flow on infinitely wide neural networks with a certain parameterization gives rise to linearized dynamics in parameter space (Lee et al., 2019; Liu et al., 2020) and consequently a kernel regression solution with a kernel known as the neural tangent kernel (NTK) in function space (Jacot et al., 2018; Arora et al., 2019). Kernel machines enjoy firmer theoretical footing than deep neural networks, which allows one to accurately study their training and generalization (Rasmussen & Williams, 2006; Schölkopf & Smola, 2002). Moreover, they share many of the phenomena that overparameterized neural networks exhibit, such as interpolating the training data (Zhang et al., 2017; Liang & Rakhlin, 2018; Belkin et al., 2018). However, the exact equivalence between neural networks and kernel machines breaks for finite width networks. Further, the regime with approximately static kernel, also referred to as the lazy training regime (Chizat et al., 2019), cannot account for the ability of deep networks to adapt their internal representations to the structure of the data, a phenomenon widely believed to be crucial to their success.

In this present study, we pursue an alternative perspective on the NTK, and ask whether a neural network with an NTK that changes significantly during training can ever be a kernel machine for a *data-dependent* kernel: i.e. does there exist a kernel function $K$ for which the final neural network function $f$ is $f(\boldsymbol{x}) \approx \sum_{\mu=1}^{P} \alpha^{\mu} K(\boldsymbol{x}, \boldsymbol{x}^{\mu})$ with coefficients $\alpha^{\mu}$ that depend only on the training data? We answer in the affirmative: that a large class of neural networks at small initialization trained on approximately whitened data are accurately approximated as kernel regression solutions with their final, data-dependent NTKs up to an error dependent on initialization scale. Hence, our results provide a further concrete link between kernel machines and deep learning which, unlike the infinite width limit, allows for the kernel to be shaped by the data.

---

\*These authors contributed equally.

The phenomenon we study consists of two training phases. In the first phase, the kernel starts off small in overall scale and quickly aligns its eigenvectors toward task-relevant directions. In the second phase, the kernel increases in overall scale, causing the network to learn a kernel regression solution with the final NTK. We call this phenomenon the *silent alignment effect* because the feature learning happens before the loss appreciably decreases. Our contributions are the following

1. In Section 2, we demonstrate the silent alignment effect by considering a simplified model where the kernel evolves while small and then subsequently increases only in scale. We theoretically show that if these conditions are met, the final neural network is a kernel machine that uses the final, data-dependent NTK. A proof is provided in Appendix B.

2. In Section 3, we provide an analysis of the NTK evolution of two layer linear MLPs with scalar target function with small initialization. If the input training data is whitened, the kernel aligns its eigenvectors towards the direction of the optimal linear function early on during training while the loss does not decrease appreciably. After this, the kernel changes in scale only, showing this setup satisfies the requirements for silent alignment discussed in Section 2.

3. In Section 4, we extend our analysis to deep MLPs by showing that the time required for alignment scales with initialization the same way as the time for the loss to decrease appreciably. Still, these time scales can be sufficiently separated to lead to the silent alignment effect for which we provide empirical evidence. We further present an explicit formula for the final kernel in linear networks of any depth and width when trained from small initialization, showing that the final NTK aligns to task-relevant directions.

4. In Section 5, we show empirically that the silent alignment phenomenon carries over to nonlinear networks trained with ReLU and Tanh activations on isotropic data, as well as linear and non-linear networks with multiple output classes. For anisotropic data, we show that the NTK must necessarily change its eigenvectors when the loss is significantly decreasing, destroying the silent alignment phenomenon. In these cases, the final neural network output deviates from a kernel machine that uses the final NTK.

## 1.1 RELATED WORKS

Jacot et al. (2018) demonstrated that infinitely wide neural networks with an appropriate parameterization trained on mean square error loss evolve their predictions as a linear dynamical system with the NTK at initalization. A limitation of this kernel regime is that the neural network internal representations and the kernel function do not evolve during training. Conditions under which such lazy training can happen is studied further in (Chizat et al., 2019; Liu et al., 2020). Domingos (2020) recently showed that every model, including neural networks, trained with gradient descent leads to a kernel model with a path kernel and coefficients $\alpha^{\mu}$ that depend on the test point $\boldsymbol{x}$. This dependence on $\boldsymbol{x}$ makes the construction not a kernel method in the traditional sense that we pursue here (see Remark 1 in (Domingos, 2020)).

Phenomenological studies and models of kernel evolution have been recently invoked to gain insight into the difference between lazy and feature learning regimes of neural networks. These include analysis of NTK dynamics which revealed that the NTK in the feature learning regime aligns its eigenvectors to the labels throughout training, causing non-linear prediction dynamics (Fort et al., 2020; Baratin et al., 2021; Shan & Bordelon, 2021; Woodworth et al., 2020; Chen et al., 2020; Geiger et al., 2021; Bai et al., 2020). Experiments have shown that lazy learning can be faster but less robust than feature learning (Flesch et al., 2021) and that the generalization advantage that feature learning provides to the final predictor is heavily task and architecture dependent (Lee et al., 2020). Fort et al. (2020) found that networks can undergo a rapid change of kernel early on in training after which the network's output function is well-approximated by a kernel method with a data-dependent NTK. Our findings are consistent with these results.

Stöger & Soltanolkotabi (2021) recently obtained a similar multiple-phase training dynamics involving an early alignment phase followed by spectral learning and refinement phases in the setting of low-rank matrix recovery. Their results share qualitative similarities with our analysis of deep linear networks. The second phase after alignment, where the kernel's eigenspectrum grows, was studied in linear networks in (Jacot et al., 2021), where it is referred to as the saddle-to-saddle regime.

Unlike prior works (Dyer & Gur-Ari, 2020; Aitken & Gur-Ari, 2020; Andreassen & Dyer, 2020), our results do not rely on perturbative expansions in network width. Also unlike the work of Saxe et al. (2014), our solutions for the evolution of the kernel do not depend on choosing a specific set of initial conditions, but rather follow only from assumptions of small initialization and whitened data.

## 2 THE SILENT ALIGNMENT EFFECT AND APPROXIMATE KERNEL SOLUTION

Neural networks in the overparameterized regime can find many interpolators: the precise function that the network converges to is controlled by the time evolution of the NTK. As a concrete example, we will consider learning a scalar target function with mean square error loss through gradient flow. Let $\boldsymbol{x} \in \mathbb{R}^D$ represent an arbitrary input to the network $f(\boldsymbol{x})$ and let $\{\boldsymbol{x}^\mu, y^\mu\}_{\mu=1}^P$ be a supervised learning training set. Under gradient flow the parameters $\boldsymbol{\theta}$ of the neural network will evolve, so the output function is time-dependent and we write this as $f(\boldsymbol{x}, t)$. The evolution for the predictions of the network on a test point can be written in terms of the NTK $K(\boldsymbol{x}, \boldsymbol{x}', t) = \frac{\partial f(\boldsymbol{x},t)}{\partial \boldsymbol{\theta}} \cdot \frac{\partial f(\boldsymbol{x}',t)}{\partial \boldsymbol{\theta}}$ as

$$\frac{d}{dt} f(\boldsymbol{x}, t) = \eta \sum_\mu K(\boldsymbol{x}, \boldsymbol{x}^\mu, t)(y^\mu - f(\boldsymbol{x}^\mu, t)), \tag{1}$$

where $\eta$ is the learning rate. If one had access to the dynamics of $K(\boldsymbol{x}, \boldsymbol{x}^\mu, t)$ throughout all $t$, one could solve for the final learned function $f^*$ with integrating factors under conditions discussed in Appendix A

$$f^*(\boldsymbol{x}) = f_0(\boldsymbol{x}) + \sum_{\mu\nu} \int_0^\infty dt \, \boldsymbol{k}_t(\boldsymbol{x})^\mu \left[ \exp\left( -\eta \int_0^t \boldsymbol{K}_{t'} \, dt' \right) \right]_{\mu\nu} (y^\nu - f_0(\boldsymbol{x}^\nu)). \tag{2}$$

Here, $\boldsymbol{k}_t(\boldsymbol{x})^\mu = K(\boldsymbol{x}, \boldsymbol{x}^\mu, t)$, $[\boldsymbol{K}_t]_{\mu,\nu} = K(\boldsymbol{x}^\mu, \boldsymbol{x}^\nu, t)$, and $y^\mu - f_0(\boldsymbol{x}^\mu)$ is the initial error on point $\boldsymbol{x}^\mu$. We see that the final function has contributions throughout the full training interval $t \in (0, \infty)$. The seminal work by Jacot et al. (2018) considers an infinite-width limit of neural networks, where the kernel function $K_t(\boldsymbol{x}, \boldsymbol{x}')$ stays constant throughout training time. In this setting where the kernel is constant and $f_0(\boldsymbol{x}^\mu) \approx 0$, then we obtain a true kernel regression solution $f(\boldsymbol{x}) = \sum_{\mu,\nu} \boldsymbol{k}(\boldsymbol{x})^\mu \boldsymbol{K}_{\mu\nu}^{-1} y^\nu$ for a kernel $K(\boldsymbol{x}, \boldsymbol{x}')$ which does not depend on the training data.

Much less is known about what happens in the rich, feature learning regime of neural networks, where the kernel evolves significantly during time in a data-dependent manner. In this paper, we consider a setting where the initial kernel is small in scale, aligns its eigenfunctions early on during gradient descent, and then increases only in scale monotonically. As a concrete phenomenological model, consider depth $L$ networks with homogenous activation functions with weights initialized with variance $\sigma^2$. At initialization $K_0(\boldsymbol{x}, \boldsymbol{x}') \sim O(\sigma^{2L-2})$, $f_0(\boldsymbol{x}) \sim O(\sigma^L)$ (see Appendix B). We further assume that after time $\tau$, the kernel only evolves in scale in a constant direction

$$K(\boldsymbol{x}, \boldsymbol{x}', t) = \begin{cases} \sigma^{2L-2} \tilde{K}(\boldsymbol{x}, \boldsymbol{x}', t) & t \leq \tau \\ g(t) K_\infty(\boldsymbol{x}, \boldsymbol{x}') & t > \tau \end{cases}, \tag{3}$$

where $\tilde{K}(\boldsymbol{x}, \boldsymbol{x}', t)$ evolves from an initial kernel at time $t = 0$ to $K_\infty(\boldsymbol{x}, \boldsymbol{x}')$ by $t = \tau$ and $g(t)$ increases monotonically from $\sigma^{2L-2}$ to $1$. In this model, one also obtains a kernel regression solution in the limit where $\sigma \to 0$ with the final, rather than the initial kernel: $f(\boldsymbol{x}) = \boldsymbol{k}_\infty(\boldsymbol{x}) \cdot \boldsymbol{K}_\infty^{-1} \boldsymbol{y} + O(\sigma^L)$. We provide a proof of this in the Appendix B.

The assumption that the kernel evolves early on in gradient descent before increasing only in scale may seem overly strict as a model of kernel evolution. However, we analytically show in Sections 3 and 4 that this can happen in deep linear networks initialized with small weights, and consequently that the final learned function is a kernel regression with the final NTK. Moreover, we show that for a linear network with small weight initialization, the final NTK depends on the training data in a universal and predictable way.

We show empirically that our results carry over to nonlinear networks with ReLU and tanh activations under the condition that the data is whitened. For example, see Figure 1, where we show the silent alignment effect on ReLU networks with whitened MNIST and CIFAR-10 images. We define alignment as the overlap between the kernel and the target function $\frac{\boldsymbol{y}^\top \boldsymbol{K} \boldsymbol{y}}{\|\boldsymbol{K}\|_F |\boldsymbol{y}|^2}$, where $\boldsymbol{y} \in \mathbb{R}^P$ is

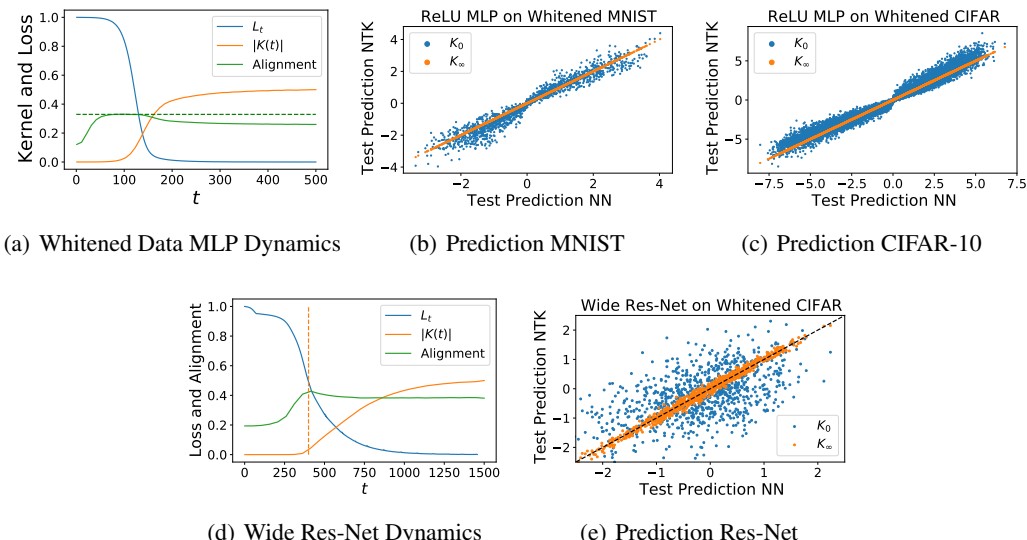

(a) Whitened Data MLP Dynamics     (b) Prediction MNIST     (c) Prediction CIFAR-10

(d) Wide Res-Net Dynamics     (e) Prediction Res-Net

Figure 1: A demonstration of the Silent Alignment effect. (a) We trained a 2-layer ReLU MLP on $P = 1000$ MNIST images of handwritten 0's and 1's which were whitened. Early in training, around $t \approx 50$, the NTK aligns to the target function and stay fixed (green). The kernel's overall scale (orange) and the loss (blue) begin to move at around $t = 300$. The analytic solution for the maximal final alignment value in linear networks is overlayed (dashed green), see Appendix E.2. (b) We compare the predictions of the NTK and the trained network on MNIST test points. Due to silent alignment, the final learned function is well described as a kernel regression solution with the final NTK $K_\infty$. However, regression with the initial NTK is not a good model of the network's predictions. (c) The same experiment on $P = 1000$ whitened CIFAR-10 images from the first two classes. Here we use MSE loss on a width 100 network with initialization scale $\sigma = 0.1$. (d) Wide-ResNet with width multiplier $k = 4$ and blocksize of $b = 1$ trained with $P = 100$ training points from the first two classes of CIFAR-10. The dashed orange line marks when the kernel starts growing significantly, by which point the alignment has already finished. (e) Predictions of the final NTK are strongly correlated with the final NN function.

a vector of the target values, quantifying the projection of the labels onto the kernel, as discussed in (Cortes et al., 2012). This quantity increases early in training but quickly stabilizes around its asymptotic value before the loss decreases. Though Equation 2 was derived under assumption of gradient flow with constant learning rate, the underlying conclusions can hold in more realistic settings as well. In Figure 1 (d) and (e) we show learning dynamics and network predictions for Wide-ResNet (Zagoruyko & Komodakis, 2017) on whitened CIFAR-10 trained with the Adam optimizer (Kingma & Ba, 2014) with learning rate $10^{-5}$, which exhibits silent alignment and strong correlation with the final NTK predictor. In the unwhitened setting, this effect is partially degraded, as we discuss in Section 5 and Appendix J. Our results suggest that the final NTK may be useful for analyzing generalization and transfer as we discuss for the linear case in Appendix F.

## 3   KERNEL EVOLUTION IN 2 LAYER LINEAR NETWORKS

We will first study shallow linear networks trained with small initialization before providing analysis for deeper networks in Section 4. We will focus our discussion in this section on the scalar output case but we will provide similar analysis in the multiple output channel case in a subsequent section. We demonstrate that our analytic solutions match empirical simulations in Appendix C.5.

We assume the $P$ data points $\boldsymbol{x}^\mu \in \mathbb{R}^D, \mu = 1, \ldots, P$ of zero mean with correlation matrix $\boldsymbol{\Sigma} = \frac{1}{P} \sum_{\mu=1}^{P} \boldsymbol{x}^\mu \boldsymbol{x}^{\mu\top}$. Further, we assume that the target values are generated by a linear teacher function $y^\mu = s\boldsymbol{\beta}_T \cdot \boldsymbol{x}^\mu$ for a unit vector $\boldsymbol{\beta}_T$. The scalar $s$ merely quantifies the size of the supervised learning signal: the variance of $|\boldsymbol{y}|^2 = s^2 \boldsymbol{\beta}_T^\top \boldsymbol{\Sigma} \boldsymbol{\beta}_T$. We define the two-layer linear neu-

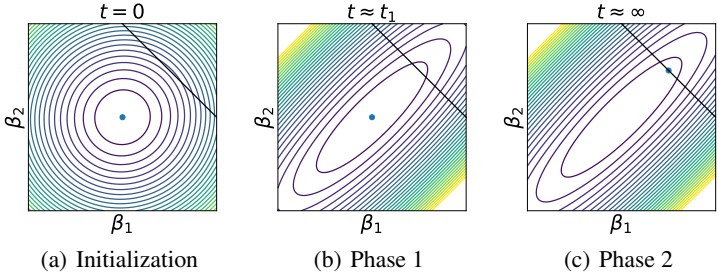

|           |           |           |
|-----------|-----------|-----------|
| (a) Initialization | (b) Phase 1 | (c) Phase 2 |

Figure 2: The evolution of the kernel's eigenfunctions happens during the early alignment phase for $t_1 \approx \frac{1}{s}$, but significant evolution in the network predictions happens for $t > t_2 = \frac{1}{2} \log(s\sigma^{-2})$. (a) Contour plot of kernel's norm for linear functions $f(\boldsymbol{x}) = \boldsymbol{\beta} \cdot \boldsymbol{x}$. The black line represents the space of weights which interpolate the training set, ie $\boldsymbol{X}^\top \boldsymbol{\beta} = \boldsymbol{y}$. At initialization, the kernel is isotropic, resulting in spherically symmetric level sets of RKHS norm. The network function is represented as a blue dot. (b) During Phase I, the kernel's eigenfunctions have evolved, enhancing power in the direction of the min-norm interpolator, but the network function has not moved far from the origin. (c) In Phase II, the network function $\boldsymbol{W}^\top \boldsymbol{a}$ moves from the origin to the final solution.

ral network with $N$ hidden units as $f(\boldsymbol{x}) = \boldsymbol{a}^\top \boldsymbol{W} \boldsymbol{x}$. Concretely, we initialize the weights with standard parameterization $a_i \sim \mathcal{N}(0, \sigma^2/N), W_{ij} \sim \mathcal{N}(0, \sigma^2/D)$. Understanding the role of $\sigma$ in the dynamics will be crucial to our study. We analyze gradient flow dynamics on MSE cost $L = \frac{1}{2P} \sum_\mu \left( f(\boldsymbol{x}^\mu) - y^\mu \right)^2$.

Under gradient flow with learning rate $\eta = 1$, the weight matrices in each layer evolve as

$$\frac{d}{dt}\boldsymbol{a} = -\frac{\partial L}{\partial \boldsymbol{a}} = \boldsymbol{W}\boldsymbol{\Sigma}\left(s\boldsymbol{\beta}_T - \boldsymbol{W}^\top \boldsymbol{a}\right), \quad \frac{d}{dt}\boldsymbol{W} = -\frac{\partial L}{\partial \boldsymbol{W}} = \boldsymbol{a}\left(s\boldsymbol{\beta}_T - \boldsymbol{W}^\top \boldsymbol{a}\right)^\top \boldsymbol{\Sigma}. \quad (4)$$

The NTK takes the following form throughout training.

$$K(\boldsymbol{x}, \boldsymbol{x}'; t) = \boldsymbol{x}^\top \boldsymbol{W}^\top \boldsymbol{W} \boldsymbol{x}' + |\boldsymbol{a}|^2 \boldsymbol{x}^\top \boldsymbol{x}'. \quad (5)$$

Note that while the second term, a simple isotropic linear kernel, does not reflect the nature of the learning task, the first term $\boldsymbol{x}^\top \boldsymbol{W}^\top \boldsymbol{W} \boldsymbol{x}'$ can evolve to yield an anisotropic kernel that has learned a representation from the data.

## 3.1 Phases of Training in Two Layer Linear Network

We next show that there are essentially two phases of training when training a two-layer linear network from small initialization on whitened-input data.

- Phase I: An alignment phase which occurs for $t \sim \frac{1}{s}$. In this phase the weights align to their low rank structure and the kernel picks up a rank-one term of the form $\boldsymbol{x}^\top \boldsymbol{\beta}\boldsymbol{\beta}^\top \boldsymbol{x}'$. In this setting, since the network is initialized near $\boldsymbol{W}, \boldsymbol{a} = \boldsymbol{0}$, which is a saddle point of the loss function, the gradient of the loss is small. Consequently, the magnitudes of the weights and kernel evolve slowly.
- Phase II: A data fitting phase which begins around $t \sim \frac{1}{s} \log(s\sigma^{-2})$. In this phase, the system escapes the initial saddle point $\boldsymbol{W}, \boldsymbol{a} = 0$ and loss decreases to zero. In this setting both the kernel's overall scale and the scale of the function $f(\boldsymbol{x}, t)$ increase substantially.

If Phase I and Phase II are well separated in time, which can be guaranteed by making $\sigma$ small, then the final function solves a kernel interpolation problem for the NTK which is only sensitive to the geometry of gradients in the final basin of attraction. In fact, in the linear case, the kernel interpolation at every point along the gradient descent trajectory would give the final solution as we show in Appendix G. A visual summary of these phases is provided in Figure 2.

### 3.1.1 Phase I: Early Alignment for Small Initialization

In this section we show how the kernel aligns to the correct eigenspace early in training. We focus on the whitened setting, where the data matrix $\boldsymbol{X}$ has all of its nonzero singular values equal. We let

$\boldsymbol{\beta}$ represent the normalized component of $\boldsymbol{\beta}_T$ in the span of the training data $\{\boldsymbol{x}^\mu\}$. We will discuss general $\boldsymbol{\Sigma}$ in section 3.2. We approximate the dynamics early in training by recognizing that the network output is small due to the small initialization. Early on, the dynamics are given by:

$$\frac{d}{dt}\boldsymbol{a} = s\boldsymbol{W}\boldsymbol{\beta} + O(\sigma^3)\,, \quad \frac{d}{dt}\boldsymbol{W} = s\boldsymbol{a}\boldsymbol{\beta}^\top + O(\sigma^3). \qquad (6)$$

Truncating terms order $\sigma^3$ and higher, we can solve for the kernel's dynamics early on in training

$$K(\boldsymbol{x}, \boldsymbol{x}'; t) = q_0 \cosh(2\eta st)\, \boldsymbol{x}^\top \left[\boldsymbol{\beta}\boldsymbol{\beta}^\top + \boldsymbol{I}\right] \boldsymbol{x}' + O(\sigma^2), \quad t \ll s^{-1} \log(s/\sigma^2). \qquad (7)$$

where $q_0$ is an initialization dependent quantity, see Appendix C.1. The bound on the error is obtained in Appendix C.2. We see that the kernel picks up a rank one-correction $\boldsymbol{\beta}\boldsymbol{\beta}^\top$ which points in the direction of the task vector $\boldsymbol{\beta}$, indicating that the kernel evolves in a direction sensitive to the target function $y = s\boldsymbol{\beta}_T \cdot \boldsymbol{x}$. This term grows exponentially during the early stages of training, and overwhelms the original kernel $K_0$ with timescale $1/s$. Though the neural network has not yet achieved low loss in this phase, the alignment of the kernel and learned representation has consequences for the transfer ability of the network on correlated tasks as we show in Appendix F.

### 3.1.2 PHASE II: SPECTRAL LEARNING

We now assume that the weights have approached their low rank structure, as predicted from the previous analysis of Phase I dynamics, and study the subsequent NTK evolution. We will show that, under the assumption of whitening, the kernel only evolves in overall scale.

First, following (Fukumizu, 1998; Arora et al., 2018; Du et al., 2018), we note the following conservation law $\frac{d}{dt}\left[\boldsymbol{a}(t)\boldsymbol{a}(t)^\top - \boldsymbol{W}(t)\boldsymbol{W}(t)^\top\right] = 0$ which holds for all time. If we assume small initial weight variance $\sigma^2$, $\boldsymbol{a}\boldsymbol{a}^\top - \boldsymbol{W}\boldsymbol{W}^\top = O(\sigma^2) \approx 0$ at initialization, and stays that way during the training due to the conservation law. This condition is surprisingly informative, since it indicates that $\boldsymbol{W}$ is rank-one up to $O(\sigma)$ corrections. From the analysis of the alignment phase, we also have that $\boldsymbol{W}^\top\boldsymbol{W} \propto \boldsymbol{\beta}\boldsymbol{\beta}^\top$. These two observations uniquely determine the rank one structure of $\boldsymbol{W}$ to be $\boldsymbol{a}\boldsymbol{\beta}^\top + O(\sigma)$. Thus, from equation 5 it follows that in Phase II, the kernel evolution takes the form

$$K(\boldsymbol{x}, \boldsymbol{x}'; t) = u(t)^2 \boldsymbol{x}^\top \left[\boldsymbol{\beta}\boldsymbol{\beta}^\top + \boldsymbol{I}\right] \boldsymbol{x}' + O(\sigma), \qquad (8)$$

where $u(t)^2 = |\boldsymbol{a}|^2$. This demonstrates that the kernel only changes in overall scale during Phase II.

Once the weights are aligned with this scheme, we can get an expression for the evolution of $u(t)^2$ analytically, $u(t)^2 = se^{2st}(e^{2st} - 1 + s/u_0^2)^{-1}$, using the results of (Fukumizu, 1998; Saxe et al., 2014) as we discuss in C.4. This is a sigmoidal curve which starts at $u_0^2$ and approaches $s$. The transition time where active learning begins occurs when $e^{st} \approx s/u_0^2 \implies t \approx s^{-1} \log(s/\sigma^2)$. This analysis demonstrates that the kernel only evolves in scale during this second phase in training from the small initial value $u_0^2 \sim O(\sigma^2)$ to its asymptote.

Hence, kernel evolution in this scenario is equivalent to the assumptions discussed in Section 2, with $g(t) = u(t)^2$, showing that the final solution is well approximated by kernel regression with the final NTK. We stress that the timescale for the first phase $t_1 \sim 1/s$, where eigenvectors evolve, is independent of the scale of the initialization $\sigma^2$, whereas the second phase occurs around $t_2 \approx t_1 \log(s/\sigma^2)$. This separation of timescales $t_1 \ll t_2$ for small $\sigma$ guarantees the silent alignment effect. We illustrate these learning curves and for varying $\sigma$ in Figure C.2.

### 3.2 UNWHITENED DATA

When data is unwhitened, the right singular vector of $\boldsymbol{W}$ aligns with $\boldsymbol{\Sigma}\boldsymbol{\beta}$ early in training, as we show in Appendix C.3. This happens since, early on, the dynamics for the first layer are $\frac{d}{dt}\boldsymbol{W} \sim \boldsymbol{a}(t)\boldsymbol{\beta}^\top\boldsymbol{\Sigma}$. Thus the early time kernel will have a rank-one spike in the $\boldsymbol{\Sigma}\boldsymbol{\beta}$ direction. However, this configuration is not stable as the network outputs grow. In fact, at late time $\boldsymbol{W}$ must realign to converge to $\boldsymbol{W} \propto \boldsymbol{a}\boldsymbol{\beta}^\top$ since the network function converges to the optimum and $f = \boldsymbol{a}^\top\boldsymbol{W}\boldsymbol{x} = s\boldsymbol{\beta} \cdot \boldsymbol{x}$, which is the minimum $\ell_2$ norm solution (Appendix G.1). Thus, the final kernel will always look like $K_\infty(\boldsymbol{x}, \boldsymbol{x}') = s\boldsymbol{x}^\top \left[\boldsymbol{\beta}\boldsymbol{\beta}^\top + \boldsymbol{I}\right] \boldsymbol{x}'$. However, since the realignment of $\boldsymbol{W}$'s singular vectors happens *during the Phase II spectral learning*, the kernel is not constant up to overall scale, violating the conditions for silent alignment. We note that the learned function still is a kernel regression solution of the final NTK, which is a peculiarity of the linear network case, but this is not achieved through the silent alignment phenomenon as we explain in Appendix C.3.

## 4    EXTENSION TO DEEP LINEAR NETWORKS

We next consider scalar target functions approximated by deep linear neural networks and show that many of the insights from the two layer network carry over. The neural network function $f : \mathbb{R}^D \to \mathbb{R}$ takes the form $f(\boldsymbol{x}) = \boldsymbol{w}^{L\top} \boldsymbol{W}^{L-1}...\boldsymbol{W}^1 \boldsymbol{x}$. The gradient flow dynamics under mean squared error (MSE) loss become

$$\frac{d}{dt}\boldsymbol{W}^\ell = -\eta \frac{\partial L}{\partial \boldsymbol{W}^\ell} = \eta \left(\prod_{\ell' > \ell} \boldsymbol{W}^{\ell'}\right)^\top (s\boldsymbol{\beta} - \tilde{\boldsymbol{w}})^\top \boldsymbol{\Sigma} \left(\prod_{\ell' < \ell} \boldsymbol{W}^{\ell'}\right)^\top, \tag{9}$$

where $\tilde{\boldsymbol{w}} = \boldsymbol{W}^{1\top}\boldsymbol{W}^{2\top}...\boldsymbol{w}^L \in \mathbb{R}^D$ is shorthand for the effective one-layer linear network weights. Inspired by observations made in prior works (Fukumizu, 1998; Arora et al., 2018; Du et al., 2018), we again note that the following set of conservation laws hold during the dynamics of gradient descent $\frac{d}{dt}\left[\boldsymbol{W}^\ell \boldsymbol{W}^{\ell\top} - \boldsymbol{W}^{\ell+1\top}\boldsymbol{W}^{\ell+1}\right] = 0$. This condition indicates a balance in the size of weight updates in adjacent layers and simplifies the analysis of linear networks. This balancing condition between weights of adjacent layers is not specific to MSE loss, but will also hold for any loss function, see Appendix D. We will use this condition to characterize the NTK's evolution.

### 4.1    NTK UNDER SMALL INITIALIZATION

We now consider the effects of small initialization. When the initial weight variance $\sigma^2$ is sufficiently small, $\boldsymbol{W}^\ell \boldsymbol{W}^{\ell\top} - \boldsymbol{W}^{\ell+1\top}\boldsymbol{W}^{\ell+1} = O(\sigma^2) \approx 0$ at initialization.[1] This conservation law implies that these matrices remain approximately equal throughout training. Performing an SVD on each matrix and inductively using the above formula from the last layer to the first, we find that all matrices will be approximately rank-one $\boldsymbol{w}^L = u(t)\boldsymbol{r}_L(t)$ , $\boldsymbol{W}^\ell = u(t)\boldsymbol{r}_{\ell+1}(t)\boldsymbol{r}_\ell(t)^\top$, where $\boldsymbol{r}_\ell(t)$ are unit vectors. Using only this balancing condition and expanding to leading order in $\sigma$, we find that the NTK's dynamics look like

$$K(\boldsymbol{x}, \boldsymbol{x}', t) = u(t)^{2(L-1)}\boldsymbol{x}^\top \left[(L-1)\boldsymbol{r}_1(t)\boldsymbol{r}_1(t)^\top + \boldsymbol{I}\right]\boldsymbol{x}' + O(\sigma). \tag{10}$$

We derive this formula in the Appendix E. We observe that the NTK consists of a rank-1 correction to the isotropic linear kernel $\boldsymbol{x} \cdot \boldsymbol{x}'$ with the rank-one spike pointing along the $\boldsymbol{r}_1(t)$ direction. This is true dynamically throughout training under the assumption of small $\sigma$. At convergence $\boldsymbol{r}(t) \to \boldsymbol{\beta}$, which is the unique fixed point reachable through gradient descent. We discuss evolution of $u(t)$ below. The alignment of the NTK with the direction $\boldsymbol{\beta}$ increases with depth $L$.

### 4.1.1    WHITENED DATA VS ANISOTROPIC DATA

We now argue that in the case where the input data is whitened, the trained network function is again a kernel machine that uses the final NTK. The unit vector $\boldsymbol{r}_1(t)$ quickly aligns to $\boldsymbol{\beta}$ since the first layer weight matrix evolves in the rank-one direction $\frac{d}{dt}\boldsymbol{W}^1 = \boldsymbol{v}(t)\boldsymbol{\beta}^\top$ throughout training for a time dependent vector function $\boldsymbol{v}(t)$. As a consequence, early in training the top eigenvector of the NTK aligns to $\boldsymbol{\beta}$. Due to gradient descent dynamics, $\boldsymbol{W}^{1\top}\boldsymbol{W}^1$ grows only in the $\boldsymbol{\beta}\boldsymbol{\beta}^\top$ direction. Since the $\boldsymbol{r}_1$ quickly aligns to $\boldsymbol{\beta}$ due to $\boldsymbol{W}^1$ growing only along the $\boldsymbol{\beta}$ direction, then the global scalar function $c(t) = u(t)^L$ satisfies the dynamics $\dot{c}(t) = c(t)^{2-2/L}[s - c(t)]$ in the whitened data case, which is consistent with the dynamics obtained when starting from the orthogonal initialization scheme of Saxe et al. (2014). We show in the Appendix E.1 that spectral learning occurs over a timescale on the order of $t_{1/2} \approx \frac{L}{s(L-2)}\sigma^{-L+2}$, where $t_{1/2}$ is the time required to reach half the value of the initial loss. We discuss this scaling in detail in Figure 3, showing that although the timescale of alignment shares the same scaling with $\sigma$ for $L > 2$, empirically alignment in deep networks occurs faster than spectral learning. Hence, the silent alignment conditions of Section 2 are satisfied. In the case where the data is unwhitened, the $\boldsymbol{r}_1(t)$ vector aligns with $\boldsymbol{\Sigma}\boldsymbol{\beta}$ early in training. This happens since, early on, the dynamics for the first layer are $\frac{d}{dt}\boldsymbol{W}^1 \sim \boldsymbol{v}(t)\boldsymbol{\beta}^\top\boldsymbol{\Sigma}$ for time dependent vector $\boldsymbol{v}(t)$. However, for the same reasons we discussed in Section 3.2 the kernel must realign at late times, violating the conditions for silent alignment.

---

[1]Though we focus on neglecting the $O(\sigma^2)$ initial weight matrices in the main text, an approximate analysis for wide networks at finite $\sigma^2$ and large width is provided in Appendix H.2, which reveals additional dependence on relative layer widths.

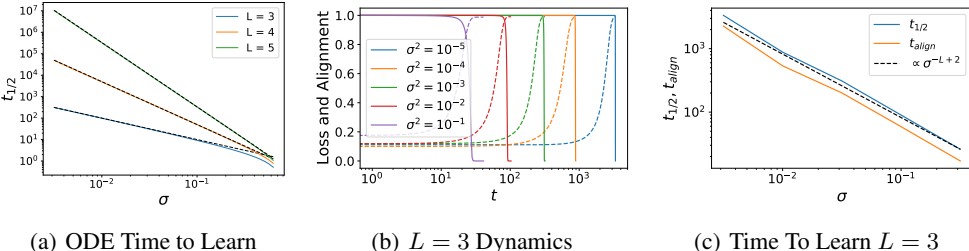

(a) ODE Time to Learn $\qquad$ (b) $L = 3$ Dynamics $\qquad$ (c) Time To Learn $L = 3$

Figure 3: (a) Time to half loss scales in a power law with $\sigma$ for networks with $L \geq 3$: $t_{1/2} \sim \frac{L}{(L-2)}\sigma^{-L+2}$ (black dashed) is compared with numerically integrating the dynamics $\dot{c}(t) = c^{2-2/L}(s - c)$ (solid). The power law scaling of $t_{1/2}$ with $\sigma$ is qualitatively different than what happens for $L = 2$, where we identified logarithmic scaling $t_{1/2} \sim \log(\sigma^{-2})$. (b) Linear networks with $D = 30$ inputs and $N = 50$ hidden units trained on synthetic whitened data with $|\boldsymbol{\beta}| = 1$. We show for a $L = 3$ linear network the cosine similarity of $\boldsymbol{W}^{1\top}\boldsymbol{W}^1$ with $\boldsymbol{\beta}\boldsymbol{\beta}^\top$ (dashed) and the loss (solid) for different initialization scales. (c) The time to get to $1/2$ the initial loss and the time for the cosine similarity of $\boldsymbol{W}^{1\top}\boldsymbol{W}^1$ with $\boldsymbol{\beta}\boldsymbol{\beta}^\top$ to reach $1/2$ both scale as $\sigma^{-L+2}$, however one can see that alignment occurs before half loss is achieved.

## 4.2 MULTIPLE OUTPUT CHANNELS

We next discuss the case where the network has multiple $C$ output channels. Each network output, we denote as $f_c(\boldsymbol{x}')$ resulting in $C^2$ kernel sub-blocks $K_{c,c'}(\boldsymbol{x}, \boldsymbol{x}') = \nabla f_c(\boldsymbol{x}) \cdot \nabla f_{c'}(\boldsymbol{x}')$. In this context, the balanced condition $\boldsymbol{W}^\ell\boldsymbol{W}^{\ell\top} \approx \boldsymbol{W}^{\ell+1\top}\boldsymbol{W}^{\ell+1}$ implies that each of the weight matrices is rank-$C$, implying a rank-$C$ kernel. We give an explicit formula for this kernel in Appendix H. For concreteness, consider whitened input data $\boldsymbol{\Sigma} = \boldsymbol{I}$ and a teacher with weights $\boldsymbol{\beta} \in \mathbb{R}^{C \times D}$. The singular value decomposition of the teacher weights $\boldsymbol{\beta} = \sum_\alpha s_\alpha \boldsymbol{z}_\alpha \boldsymbol{v}_\alpha^\top$ determines the evolution of each mode (Saxe et al., 2014). Each singular mode begins to be learned at $t_\alpha = \frac{1}{s_\alpha}\log\left(s_\alpha u_0^{-2}\right)$. To guarantee silent alignment, we need all of the Phase I time constants to be smaller than all of the Phase II time constants. In the case of a two layer network, this is equivalent to the condition $\frac{1}{s_{min}} \ll \frac{1}{s_{max}}\log\left(s_{max}u_0^{-2}\right)$ so that the kernel alignment timescales are well separated from the timescales of spectral learning. We see that alignment precedes learning in Figure H.1 (a). For deeper networks, as discussed in 4.1.1, alignment scales in the same way as the time for learning.

## 5 SILENT ALIGNMENT ON REAL DATA AND ReLU NETS

In this section, we empirically demonstrate that many of the phenomena described in the previous sections carry over to the nonlinear homogenous networks with small initialization provided that the data is not highly anisotropic. A similar separation in timescales is expected in the nonlinear $L$-homogenous case since, early in training, the kernel evolves more quickly than the network predictions. This argument is based on a phenomenon discussed by Chizat et al. (2019). Consider an initial scaling of the parameters by $\sigma$. We find that the relative change in the loss compared to the relative change in the features has the form $\frac{|\frac{d}{dt}\nabla f|}{|\nabla f|}\frac{\mathcal{L}}{|\frac{d}{dt}\mathcal{L}|} \approx O(\sigma^{-L})$ which becomes very large for small initialization $\sigma$ as we show in Appendix I. This indicates, that from small initialization, the parameter gradients and NTK evolve much more quickly than the loss. This is a necessary, but not sufficient condition for the silent alignment effect. To guarantee the silent alignment, the gradients must be finished evolving except for overall scale by the time the loss appreciably decreases. However, we showed that for whitened data that nonlinear ReLU networks do in fact enjoy the separation of timescales necessary for the silent alignment effect in Figure 1. In even more realistic settings, like ResNet in Figure 1 (d), we also see signatures of the silent alignment effect since the kernel does not grow in magnitude until the alignment has stabilized.

We now explore how anisotropic data can interfere with silent alignment. We consider the partial whitening transformation: let the singular value decomposition of the data matrix be $\boldsymbol{X} = \boldsymbol{U}\boldsymbol{S}\boldsymbol{V}^\top$ and construct a new partially whitened dataset $\boldsymbol{X}_\gamma = \boldsymbol{U}\boldsymbol{S}^\gamma\boldsymbol{V}^\top$, where $\gamma \in (0, 1)$. As $\gamma \to 0$

the dataset becomes closer to perfectly whitened. We compute loss and kernel aligment for depth 2 ReLU MLPs on a subset of CIFAR-10 and show results in Figure 4. As $\gamma \to 0$ the agreement between the final NTK and the learned neural network function becomes much closer, since the kernel alignment curve is stable after a smaller number of training steps. As the data becomes more anisotropic, the kernel's dynamics become less trivial at later time: rather than evolving only in scale, the alignment with the target function varies in a non-trivial way while the loss is decreasing. As a consequence, the NN function deviates from a kernel machine with the final NTK.

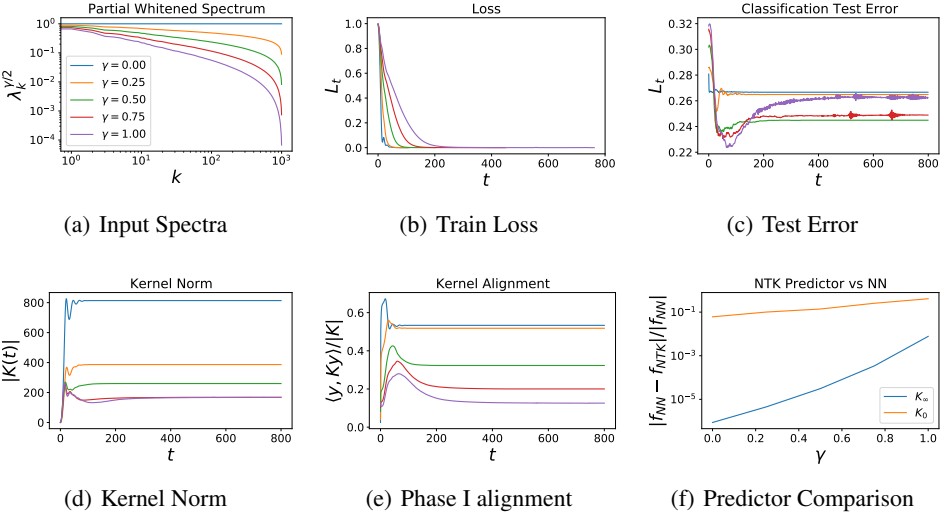

(a) Input Spectra      (b) Train Loss      (c) Test Error

(d) Kernel Norm      (e) Phase I alignment      (f) Predictor Comparison

Figure 4: Anisotropy in the data introduces multiple timescales which can interfere with the silent alignment effect in a ReLU network. Here we train an MLP to do two-class regression using Adam at learning rate $5 \times 10^{-3}$. (a) We consider the partial whitening transformation on the 1000 CIFAR-10 images $\lambda_k \to \lambda_k^\gamma$ for $\gamma \in (0, 1)$ for covariance eigenvalues $\mathbf{\Sigma} v_k = \lambda_k v_k$. (b) The loss dynamics for unwhitened data have a multitude of timescales rather than a single sigmoidal learning curve. As a consequence, kernel alignment does not happen all at once before the loss decreases and the final solution is not a kernel machine with the final NTK. (c) The network's test error on classification. (d) Anisotropic data gives a slower evolution in the kernel's Frobenius norm. (e) The kernel alignment very rapidly approaches an asymptote for whitened data but exhibits a longer timescale for the anisotropic data. (f) The final NTK predictor gives a better predictor for the neural network when the data is whitened, but still substantially outperforms the initial kernel even in the anisotropic case.

## 6 CONCLUSION

We provided an example of a case where neural networks can learn a kernel regression solution while in the rich regime. Our silent alignment phenomenon requires a separation of timescales between the evolution of the NTK's eigenfunctions and relative eigenvalues and a separate phase where the NTK grows only in scale. We demonstrate that, if these conditions are satisfied, then the final neural network function satisfies a representer theorem for the final NTK. We show analytically that these assumptions are realized in linear neural networks with small initialization trained on approximately whitened data and observe that the results hold for nonlinear networks and networks with multiple outputs. We demonstrate that silent alignment is highly sensitive to anisotropy in the input data.

Our results demonstrate that representation learning is not at odds with the learned neural network function being a kernel regression solution; i.e. a superposition of a kernel function on the training data. While we provide one mechanism for a richly trained neural network to learn a kernel regression solution through the silent alignment effect, perhaps other temporal dynamics of the NTK could also give rise to the neural network learning a kernel machine for a data-dependent kernel. Further, by asking whether neural networks behave as kernel machines for some data-dependent kernel, one can hopefully shed light on their generalization and transfer learning capabilities (Bordelon et al., 2020; Canatar et al., 2021; Loureiro et al., 2021; Geiger et al., 2021) and see Appendix F.

ACKNOWLEDGMENTS

CP acknowledges support from the Harvard Data Science Initiative. AA acknowledges support from an NDSEG Fellowship and a Hertz Fellowship. BB acknowledges the support of the NSF-Simons Center for Mathematical and Statistical Analysis of Biology at Harvard (award #1764269) and the Harvard Q-Bio Initiative. We thank Jacob Zavatone-Veth and Abdul Canatar for helpful discussions and feedback.

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

# Appendix

## A   DERIVATION OF EQUATION 2

### A.1   TRAINING POINT PREDICTIONS WITH TIME VARYING KERNEL

Given a training set of $P$ data points $\{(\boldsymbol{x}^\mu, y^\mu)\}_{\mu=1}^P$, the dynamics of the network training errors $[\boldsymbol{\Delta}_t]^\mu := f(\boldsymbol{x}^\mu, t) - y^\mu$ close in terms of a time-varying neural tangent kernel $[\boldsymbol{K}_t]_{\mu\nu} = K(\boldsymbol{x}^\mu, \boldsymbol{x}^\nu, t)$

$$\frac{d}{dt}\boldsymbol{\Delta}_t = -\boldsymbol{K}_t\boldsymbol{\Delta}_t. \tag{11}$$

We introduce the transition matrix $\boldsymbol{\Phi}_t \in \mathbb{R}^{P \times P}$ which has the property that $\boldsymbol{\Delta}_t = \boldsymbol{\Phi}_t\boldsymbol{\Delta}_0$ and $\boldsymbol{\Phi}_0 = \boldsymbol{I}$, we obtain the matrix evolution equation $\dot{\boldsymbol{\Phi}}_t = -\boldsymbol{K}_t\boldsymbol{\Phi}_t$. This equation can be solved formally in terms of the Peano-Baker series (Baake & Schlaegel, 2011; Brockett, 2015)

$$\boldsymbol{\Phi}_t = \boldsymbol{I} - \int_0^t ds_1 \boldsymbol{K}_{s_1} + \int_0^t ds_1 \boldsymbol{K}_{s_1} \int_0^{s_1} ds_2 \boldsymbol{K}_{s_2} \tag{12}$$

$$- \int_0^t ds_1 \boldsymbol{K}_{s_1} \int_0^{s_1} ds_2 \boldsymbol{K}_{s_2} \int_0^{s_2} ds_3 \boldsymbol{K}_{s_3} + ... \tag{13}$$

which can easily be verified to solve $\frac{d}{dt}\boldsymbol{\Phi}(t) = -\boldsymbol{K}(t)\boldsymbol{\Phi}(t)$ with initial condition $\boldsymbol{\Phi}(0) = \boldsymbol{I}$. Under the condition that $\int_0^t \boldsymbol{K}(t)$ commutes with $\boldsymbol{K}(t)$, which is true in the settings of interest in this paper, specifically the setting discussed in Appendix B, we can simplify the Peano-Baker series into a simple matrix exponential

$$\boldsymbol{\Phi}_t = \boldsymbol{I} - \left[\int_0^t ds\boldsymbol{K}_s\right] + \frac{1}{2}\left[\int_0^t ds\boldsymbol{K}_s\right]^2 - \frac{1}{6}\left[\int_0^t ds\boldsymbol{K}_s\right]^3 ... + \frac{(-1)^k}{k!}\left[\int_0^t ds\boldsymbol{K}_s\right]^k + ...$$

$$= \exp\left(-\int_0^t \boldsymbol{K}_s ds\right). \tag{14}$$

Thus, under the condition that $\boldsymbol{K}_t$ commutes with $\int_0^t \boldsymbol{K}_s ds$ we can exactly solve for the training error dynamics in terms of integrating factors

$$\boldsymbol{\Delta}_t = \boldsymbol{\Phi}_t\boldsymbol{\Delta}_0 = \exp\left(-\int_0^t \boldsymbol{K}_s ds\right)\boldsymbol{\Delta}_0. \tag{15}$$

We expect this formula to hold approximately whenever the eigenvectors of $\boldsymbol{K}$ are approximately equal to the eigenvectors of $\int_0^t \boldsymbol{K}_s ds$.

### A.2   TEST POINT PREDICTIONS WITH TIME VARYING KERNEL

Given access to the value of the function on training points, one can evaluate the function on test points. We have that the evolution of the function on a test point $f(\boldsymbol{x})$ is given by

$$\frac{d}{dt}f_t(\boldsymbol{x}) = -\boldsymbol{k}_t(\boldsymbol{x})\boldsymbol{\Delta}_t, \tag{16}$$

where $[\boldsymbol{k}_t(\boldsymbol{x})]^\mu := K(\boldsymbol{x}, \boldsymbol{x}^\mu, t)$. This gives the final value of $f$ to be

$$f^*(\boldsymbol{x}) := f_\infty(\boldsymbol{x}) = f_0(\boldsymbol{x}) + \int_0^\infty dt\, \boldsymbol{k}_t(\boldsymbol{x})^\mu \left[\exp\left(-\eta\int_0^t \boldsymbol{K}_{t'}dt'\right)\right](\boldsymbol{y} - \boldsymbol{f}_0). \tag{17}$$

This is exactly equation 2.

## B  KERNEL EVOLUTION IN SCALE ONLY

We consider the model of kernel evolution introduced in Section 2 where the kernel evolves only in scale for $t > \tau(\epsilon)$ and is of small overall size for $t < \tau(\epsilon)$,

$$K(x, x', t) = \begin{cases} \epsilon K_0(x, x', t) & t \leq \tau(\epsilon) \\ g(t)K_\infty(x, x') & t > \tau(\epsilon) \end{cases}. \tag{18}$$

This model allows for alignment of the kernel while small in the time window $t \in (0, \tau)$, followed by scale growth only for $t \in (\tau, \infty)$. The time threshold $\tau$ will generally depend on the initial kernel scale $\epsilon$. For example, in depth $L$ linear MLPs, $\epsilon \sim \sigma^{2L-2}$ and $\tau \sim \sigma^{-L+2}$ with initialization scale $\sigma$ as we show in Figure E.3. We will define a differentiable function $h(t) = \int_\tau^t g(t')dt'$ so that $h'(t) = g(t)$, $h(\tau) = 0$, and $\lim_{t\to\infty} h(t) = \infty$. This last condition follows from $g$'s continuity and the assumption that $\lim_{t\to\infty} g(t) = 1$. We will first show that the final neural network has the form $f(\boldsymbol{x}) = f_0(\boldsymbol{x}) + \boldsymbol{k}_\infty(x) \cdot \boldsymbol{K}_\infty^{-1}(\boldsymbol{y} - \boldsymbol{f}_0) + O(\epsilon\tau(\epsilon))$. First, we need to calculate the errors $\boldsymbol{\Delta}(t) = \boldsymbol{y} - \boldsymbol{f}(t) \in \mathbb{R}^P$ made on the $P$ training examples. These satisfy the dynamics

$$\frac{d}{dt}\boldsymbol{\Delta}(t) = -\begin{cases} \epsilon\boldsymbol{K}_0(t)\boldsymbol{\Delta}(t) & t \leq \tau \\ h'(t)\boldsymbol{K}_\infty\boldsymbol{\Delta}(t) & t > \tau \end{cases}, \tag{19}$$

where $\boldsymbol{K}_0(t), \boldsymbol{K}_\infty \in \mathbb{R}^{P \times P}$ are $P \times P$ gram matrices; e.g. $[\boldsymbol{K}_0(t)]_{\mu\nu} = K_0(\boldsymbol{x}^\mu, \boldsymbol{x}^\nu, t)$. The vector $\boldsymbol{k}(\boldsymbol{x})$ has entries given by $[\boldsymbol{k}(\boldsymbol{x})]_\mu = K(\boldsymbol{x}, \boldsymbol{x}^\mu)$. For $t \in (0, \tau)$, the error vector follows the dynamics

$$\frac{d}{dt}\boldsymbol{\Delta}(t) = -\epsilon\boldsymbol{K}_0(t)\boldsymbol{\Delta}(t). \tag{20}$$

Introducing operator norm of a matrix, $|\boldsymbol{\Phi}|_{op} = \max_{|\boldsymbol{v}|_2=1} |\boldsymbol{\Phi}\boldsymbol{v}|_2$, we will now bound the operator norm of the change in the transition matrix $\boldsymbol{\Phi}(t)$ introduced in section A.1.

**Lemma 1.** *Let $k_0 = \max_{t\in(0,\tau)} |\boldsymbol{K}_0(t)|_{op}$ represent the maximum operator norm of $\boldsymbol{K}_0$ achieved on the interval $(0, \tau)$. Let $\boldsymbol{\Phi}(t) \in \mathbb{R}^{P \times P}$ be the transition matrix for the linear dynamics of equation (20) so that $\frac{d}{dt}\boldsymbol{\Phi}(t) = -\epsilon\boldsymbol{K}_0(t)\boldsymbol{\Phi}(t)$ and $\boldsymbol{\Phi}(0) = \boldsymbol{I}$. Then,*

$$|\boldsymbol{\Phi}(\tau) - \boldsymbol{\Phi}(0)|_{op} < \epsilon\tau(\epsilon)k_0. \tag{21}$$

*Proof.* We begin by noting that, due to the triangle inequality,

$$|\boldsymbol{\Phi}(\tau) - \boldsymbol{\Phi}(0)|_{op} = \epsilon\left|\int_0^\tau \boldsymbol{K}_0(t)\boldsymbol{\Phi}(t)dt\right|_{op} \leq \epsilon\int_0^\tau |\boldsymbol{K}_0(t)\boldsymbol{\Phi}(t)|_{op}\, dt$$
$$\leq \epsilon\int_0^\tau |\boldsymbol{K}_0(t)|\,|\boldsymbol{\Phi}(t)|_{op}\, dt \leq \epsilon k_0\int_0^\tau |\boldsymbol{\Phi}(t)|_{op}\, dt. \tag{22}$$

We will now establish that $|\boldsymbol{\Phi}(t)|_{op} \leq 1$. Note that for any vector $\boldsymbol{v} \in \mathbb{R}^P$ that

$$\frac{1}{2}\frac{d}{dt}|\boldsymbol{\Phi}(t)\boldsymbol{v}|_2^2 = \boldsymbol{v}^\top\boldsymbol{\Phi}^\top\dot{\boldsymbol{\Phi}}(t)\boldsymbol{v} = -\epsilon\boldsymbol{v}^\top\boldsymbol{\Phi}(t)^\top\boldsymbol{K}(t)\boldsymbol{\Phi}(t)\boldsymbol{v} \leq 0, \tag{23}$$

where the final inequality follows from the fact that $\boldsymbol{K}(t)$ is positive semidefinite for all $t$. Therefore we have shown that $|\boldsymbol{\Phi}(t)|_{op} \leq |\boldsymbol{\Phi}(0)|_{op} = |\boldsymbol{I}|_{op} = 1$. Using this inequality, we find that

$$|\boldsymbol{\Phi}(\tau) - \boldsymbol{\Phi}(0)|_{op} \leq \epsilon k_0\int_0^\tau |\boldsymbol{\Phi}(t)|_{op}\, dt \leq \epsilon k_0\tau(\epsilon). \tag{24}$$

$\square$

With the above Lemma 1, we can bound the discrepancy $\boldsymbol{\Delta}(\tau)$ and $\boldsymbol{\Delta}(0)$, namely

$$|\boldsymbol{\Delta}(\tau) - \boldsymbol{\Delta}(0)|_2 = |(\boldsymbol{\Phi}(\tau) - \boldsymbol{\Phi}(0))\boldsymbol{\Delta}(0)|_2$$
$$\leq |\boldsymbol{\Phi}(\tau) - \boldsymbol{\Phi}(0)|_{op}|\boldsymbol{\Delta}(0)|_2 \leq \epsilon k_0\tau(\epsilon)|\boldsymbol{\Delta}(0)|_2. \tag{25}$$

This inequality must therefore hold entry-wise as well, so that

$$\boldsymbol{\Delta}(\tau) = \boldsymbol{\Delta}(0) + O(\epsilon k_0\tau(\epsilon)). \tag{26}$$

We will now establish how the training predictions $\boldsymbol{\Delta}(t)$ evolve for the second interval $t \in (\tau, \infty)$.

**Lemma 2.** *Suppose that from $t \in (\tau, \infty)$ that $\mathbf{\Delta}(t)$ obeys the dynamics $\frac{d}{dt}\mathbf{\Delta}(t) = -h'(t)\mathbf{K}_\infty \mathbf{\Delta}(t)$ where $\mathbf{\Delta}(\tau)$ is as in equation 26. Then, for all $t \in (\tau, \infty)$,*

$$\mathbf{\Delta}(t) = \exp\left(-h(t)\mathbf{K}_\infty\right)\left[(\mathbf{y} - \mathbf{f}_0) + O(\epsilon \tau(\epsilon)k_0)\right]. \tag{27}$$

*Proof.* The differential equation can be solved through eigendecomposition and integrating factors. Let $\Delta_k(t)$ represent the $k$-th component of $\mathbf{\Delta}(t)$ in the eigenbasis of $\mathbf{K}_\infty$ which is static for $t \in (\tau, \infty)$. Let the corresponding eigenvalue of $\mathbf{K}_\infty$ be $\lambda_k$. The scalar variable $\Delta_k(t)$ obeys the dynamics

$$\frac{d}{dt}\Delta_k(t) = -\lambda_k h'(t)\Delta_k(t). \tag{28}$$

This can be solved with integrating factors, noting that $\frac{d}{dt}\left[e^{\lambda_k h(t)}\Delta_k(t)\right] = 0$. This implies that $\Delta_k(t) = e^{-\lambda_k h(t)}\Delta_k(\tau)$. Written as a vector, $\mathbf{\Delta}(t) = \exp\left(-h(t)\mathbf{K}_\infty\right)\mathbf{\Delta}(\tau)$. Since by Lemma 1 we have $\mathbf{\Delta}(\tau) = \mathbf{\Delta}(0) + O(\epsilon k_0 \tau(\epsilon))$, we obtain the desired result. $\qquad\square$

We will now combine the results of the previous two lemmas which analyze the evolution of the network predictions on the training set to give our main silent alignment result, which specifies what the neural network function predicts for an arbitrary test point $\mathbf{x}$.

**Theorem 1.** *Let the kernel have dynamics of Equation 18 where $g(t)$ is a continuous, integrable function with $\lim_{t\to\infty} g(t) = 1$. The function learned by the neural network is*

$$f(\mathbf{x}) - f_0(\mathbf{x}) = \mathbf{k}_\infty(\mathbf{x}) \cdot \mathbf{K}_\infty^{-1}\mathbf{y} + O_\epsilon(\epsilon \tau(\epsilon)). \tag{29}$$

*Proof.* Using Lemma 1 and 2, we know the full dynamics for training predictions $\mathbf{\Delta}(t)$. Using $\mathbf{\Delta}(t)$, we can solve for the final predictor $f(\mathbf{x})$ by integrating dynamics $\dot{f}(\mathbf{x}, t) = \mathbf{k}(\mathbf{x}, t) \cdot \mathbf{\Delta}(t)$.

$$f(\mathbf{x}) - f_0(\mathbf{x}) = \epsilon \int_0^\tau \mathbf{k}_0(\mathbf{x}, t) \cdot \mathbf{\Delta}(t)dt + \mathbf{k}_\infty(\mathbf{x}) \cdot \int_\tau^\infty h'(t)\exp\left(-h(t)\mathbf{K}_\infty\right)\mathbf{\Delta}(\tau)dt \tag{30}$$

We will now bound the first term. Taking $\tilde{k}_0 = \max_{t\in(0,\tau), \mathbf{x}\in\mathbb{R}^D} |\mathbf{k}_0(\mathbf{x}, t)|_2$, we get that

$$\left|\epsilon \int_0^\tau \mathbf{k}_0(\mathbf{x}, t) \cdot \mathbf{\Delta}(t)\right| \leq \epsilon \int_0^\tau |\mathbf{k}(\mathbf{x})||\mathbf{\Delta}(t)|dt \leq \epsilon(1 + \epsilon\tau(\epsilon)k_0)|\mathbf{\Delta}_0| \int_0^\tau |\mathbf{k}_0(\mathbf{x}, t)|dt$$

$$\leq \epsilon\tau(\epsilon)\tilde{k}_0(1 + \epsilon\tau(\epsilon)k_0)|\mathbf{\Delta}(0)| = O_\epsilon(\epsilon\tau(\epsilon)).$$

We can now integrate the matrix exponential in the second term, using the fact that

$$\int_\tau^\infty h'(t)\exp\left(-h(t)\mathbf{K}_\infty\right)dt = \int_0^\infty \exp\left(-h\mathbf{K}_\infty\right)dh = \mathbf{K}_\infty^{-1}. \tag{31}$$

Using the fact that $\mathbf{\Delta}(\tau) = \mathbf{\Delta}(0) + O(\tau(\epsilon)\epsilon)$ from Lemma 1, we arrive at the desired result

$$f(\mathbf{x}) - f_0(\mathbf{x}) = \mathbf{k}_\infty(\mathbf{x})\mathbf{K}_\infty^{-1}\mathbf{\Delta}_0 + O_\epsilon(\epsilon\tau(\epsilon)). \tag{32}$$

$\qquad\square$

We have now established that, given the kernel dynamics in Equation 18, $f(\mathbf{x}) - f_0(\mathbf{x})$ converges to the kernel regression solution with final NTK as $\epsilon \to 0$ provided $\lim_{\epsilon\to 0}\epsilon\tau(\epsilon) = 0$. This is generic in the settings we consider in this paper for networks with small initialization. In this small initialization setting, $f_0$ is also negligible so that $f(\mathbf{x})$ itself is a kernel regression solution. For example, in a linear depth $L$ neural network with initial weight scale $\sigma$, the initial scale of the kernel is $\epsilon \sim \sigma^{2L-2}$ while the time to alignment scales as $\tau \sim \sigma^{2-L}$ thus $\epsilon\tau \sim \sigma^L$ can be made arbitrarily small by taking $\sigma \to 0$. Lastly, the initial network outputs $f_0(\mathbf{x}) \sim \sigma^L$ can also be made arbitrarily small.

## C  PHASES OF LEARNING AT SMALL INITIALIZATION

### C.1  PHASE I: TWO LAYER NETWORK AND KERNEL ALIGNMENT

We now present an analysis distinct from that of the previous subsection to go beyond the first step of gradient descent. The NTK for the two layer linear network has the form $K = \boldsymbol{x}^\top \boldsymbol{M} \boldsymbol{x}'$ with $\boldsymbol{M} = \boldsymbol{W}^\top \boldsymbol{W} + |\boldsymbol{a}|^2 \boldsymbol{I}$. Our goal is to determine the eigendecomposition of $\boldsymbol{M}$. Introduce the variables $q(t) = \frac{1}{2} \boldsymbol{\beta}^\top \boldsymbol{M} \boldsymbol{\beta} = \frac{1}{2} \left[ |\boldsymbol{a}|^2 + \boldsymbol{\beta}^\top \boldsymbol{W}^\top \boldsymbol{W} \boldsymbol{\beta} \right]$ and $r(t) = \boldsymbol{a}^\top \boldsymbol{W} \boldsymbol{\beta}$. These dynamics form a closed two dimensional linear system early in training

$$
\frac{d}{dt} \begin{bmatrix} q(t) \\ r(t) \end{bmatrix} = 2\eta s \begin{bmatrix} 0 & 1 \\ 1 & 0 \end{bmatrix} \begin{bmatrix} q(t) \\ r(t) \end{bmatrix} + O(\sigma^3) , \quad t, \sigma \to 0
$$
$$
\implies \begin{bmatrix} q(t) \\ r(t) \end{bmatrix} = \frac{1}{2}(q_0 + r_0) \begin{bmatrix} 1 \\ 1 \end{bmatrix} e^{2\eta s t} + \frac{1}{2}(q_0 - r_0) \begin{bmatrix} 1 \\ -1 \end{bmatrix} e^{-2\eta s t} + O(\sigma^3)
$$
(33)

The variable $q(t)$ represents the alignment of the NTK with the optimal direction $\boldsymbol{\beta}$ while $r(t)$ defines the alignment of the network with the teacher. We see that this alignment increases exponentially with timescale $t \sim \eta^{-1} s^{-1}$. While the above equations hold for early time and small initialization for any initial condition $q_0, r_0$, we can further estimate these initial values under random initialization provided the input dimension is large. We stress that this limit is not necessary for the silent alignment, but allows for a nice simplification. For Gaussian initialization $a_i \sim \mathcal{N}(0, \sigma^2/N), W_{ij} \sim \mathcal{N}(0, \sigma^2/D)$ with large $D$, we have

$$
\langle q_0 \rangle = \frac{\sigma^2}{2} \left( 1 + \frac{N}{D} \right) , \quad \langle r_0 \rangle = 0 , \quad \langle r_0^2 \rangle = \frac{\sigma^4}{D}.
$$
(34)

In the large $D$ limit, we have with high probability $q_0 \gg r_0$ and thus $r(t) = q_0 \sinh(2\eta s t)$ and $q = q_0 \cosh(2\eta s t)$. Note that this gives the quantities $q(t) = \frac{1}{2} \boldsymbol{\beta}^\top \boldsymbol{M}(t) \boldsymbol{\beta}, r(t) = \boldsymbol{\beta}^\top \boldsymbol{W}^\top \boldsymbol{a}$ early in training. Now consider a unit vector $\boldsymbol{v}$ which is orthogonal to the solution $\boldsymbol{\beta}^\top \boldsymbol{v} = 0$. We find that the projection of $\boldsymbol{M}$ along this direction evolves dynamically as:

$$
\frac{d}{dt} \boldsymbol{v}^\top \boldsymbol{M}(t) \boldsymbol{v} = 2\boldsymbol{v}^\top \left[ \boldsymbol{\beta} \boldsymbol{a}^\top \boldsymbol{W} + \boldsymbol{\beta}^\top \boldsymbol{W}^\top \boldsymbol{a} \boldsymbol{I} \right] \boldsymbol{v}
$$
$$
= 2r(t).
$$
(35)

We can conclude that $\boldsymbol{v}^\top \boldsymbol{M} \boldsymbol{v}$ is equal to $q(t)$ up to an additive initialization constant. We see that this is evolving half as quickly as $\boldsymbol{\beta}^\top \boldsymbol{M} \boldsymbol{\beta} = 2q(t)$. Since $\boldsymbol{v}^\top \boldsymbol{M}_0 \boldsymbol{v} \sim O(\sigma^2)$ is small compared to the exponentially growing $\boldsymbol{M}(t)$, the only matrix that satisfies these two conditions must necessarily take the form

$$
\boldsymbol{M}(t) = q_0 \cosh(2\eta s t) \left[ \boldsymbol{\beta} \boldsymbol{\beta}^\top + \boldsymbol{I} \right] + \boldsymbol{M}_0.
$$
(36)

The first term, which is growing exponentially in $t$ will eventually overwhelm the randomly initialized kernel $\boldsymbol{M}_0$, which is $O(\sigma^2)$.

### C.2  PHASE I: ERROR IN THE LEADING ORDER APPROXIMATION

In solving the equations of the previous section, we truncated the full gradient descent equations at order $\sigma^3$. It is important to confirm that the error generated by this truncation remains bounded. We will argue by self-consistency. The full equations are

$$
\frac{d}{dt} \boldsymbol{a} = \boldsymbol{W} \left( s\boldsymbol{\beta} - \boldsymbol{W}^\top \boldsymbol{a} \right), \quad \frac{d}{dt} \boldsymbol{W} = \boldsymbol{a} \left( s\boldsymbol{\beta} - \boldsymbol{W}^\top \boldsymbol{a} \right)^\top.
$$
(37)

One can use these equations to solve for the dynamics of the $r, q$ variables:

$$
\frac{d}{dt} q(t) = 2sr - \boldsymbol{\beta}^\top \left[ \boldsymbol{W}^\top \boldsymbol{a} \boldsymbol{a}^\top \boldsymbol{W} + \boldsymbol{a}^\top \boldsymbol{W} \boldsymbol{W}^\top \boldsymbol{a} \right] \boldsymbol{\beta} = 2sr - [2r^2 + \sum_{\boldsymbol{v}_i \perp \boldsymbol{\beta}} (\boldsymbol{a}^\top \boldsymbol{W} \boldsymbol{v}_i)^2]
$$
(38)

$$
\frac{d}{dt} r(t) = 2sq - \boldsymbol{a}^\top [\boldsymbol{W} \boldsymbol{W}^\top + \boldsymbol{a} \boldsymbol{a}^\top] \boldsymbol{W} \boldsymbol{\beta} = 2sq - 2|\boldsymbol{a}|^2 r + \boldsymbol{a}^\top [\boldsymbol{a} \boldsymbol{a}^\top - \boldsymbol{W} \boldsymbol{W}^\top] \boldsymbol{W} \boldsymbol{\beta}
$$
(39)

The second equality in equation 38 comes from inserting a complete basis of states including $\boldsymbol{\beta}$ and $\boldsymbol{v}_i \perp \boldsymbol{\beta}$ into the last term of the left-hand side. The second equality in equation 39 comes from writing $\boldsymbol{W}\boldsymbol{W}^\top + \boldsymbol{a}\boldsymbol{a}^\top = 2\boldsymbol{a}\boldsymbol{a}^\top + (\boldsymbol{W}\boldsymbol{W}^\top - \boldsymbol{a}\boldsymbol{a}^\top)$. Note that the final term in brackets on the right hand side is a conserved quantity for linear networks, and so is always of order $O(\sigma^2)$.

Assuming the solutions for $q, r$ are valid to order $\sigma^2$, we get that $\frac{d}{dt}\boldsymbol{a}^\top\boldsymbol{W}\boldsymbol{v}_i = O(\sigma^4)$.

$$\frac{d}{dt}\boldsymbol{a}^\top\boldsymbol{W}\boldsymbol{v}_i = (\boldsymbol{\beta} - \hat{\boldsymbol{\beta}})^\top\boldsymbol{W}^\top\boldsymbol{W}\boldsymbol{v}_i + |\boldsymbol{a}|^2(\boldsymbol{\beta} - \hat{\boldsymbol{\beta}})^\top\boldsymbol{v}_i \tag{40}$$

Further noting that because of the conservation law $\boldsymbol{a}\boldsymbol{a}^\top - \boldsymbol{W}\boldsymbol{W}^\top = O(\sigma^2)$ is also constant in time. This gives us that

$$|\boldsymbol{a}^\top[\boldsymbol{a}\boldsymbol{a}^\top - \boldsymbol{W}\boldsymbol{W}^\top]\boldsymbol{W}\boldsymbol{\beta}| \leq \sigma^2|\boldsymbol{a}^\top\boldsymbol{W}\boldsymbol{\beta}| = \sigma^2 r. \tag{41}$$

We now note that $r, |\boldsymbol{a}|^2$ both grow as a ($\sigma$, $s$-independent) constant times times $\sigma^2 e^{2st}$. The correction to the dynamics of both equations is then bounded by a constant times $\sigma^4 e^{4st}$. This will be less than $\sigma^2$ as long as $t$ satisfies

$$t \ll \frac{1}{4s}\log\left(\frac{s}{\sigma^2}\right). \tag{42}$$

For $\sigma^2 \ll s$, the alignment time $t = 1/s$ falls within this range and we are guaranteed alignment to the Ganguli-Saxe configuration.

The error of the full solution at time $t$ can be bounded by the integral of this error bound from $0$ to $t$, namely a constant times $\sigma^4/s$. As long as $s \gg \sigma^4$, we are guaranteed that the error of the kernel is $O(\sigma^2)$ as given in equation 7.

## C.3 PHASE I: TWO LAYER ANALYSIS WITH UNWHITENED DATA

We now study the same linearization around the initial fixed point used in the main text but for the two layer network with unwhitened data. In this case,

$$\frac{d}{dt}\boldsymbol{a} \sim s\boldsymbol{W}\boldsymbol{\Sigma}\boldsymbol{\beta}, \tag{43}$$

$$\frac{d}{dt}\boldsymbol{W} \sim s\boldsymbol{a}\boldsymbol{\beta}^\top\boldsymbol{\Sigma}. \tag{44}$$

which holds asymptotically as $t/\log(\sigma^{-1}) \to 0$. We introduce the following variables which form a closed linear dynamical system

$$\begin{aligned}
r_1(t) &= \boldsymbol{\beta}^\top\boldsymbol{W}^\top\boldsymbol{a}, \\
r_2(t) &= |\boldsymbol{a}|^2, \\
r_3(t) &= \boldsymbol{\beta}^\top\boldsymbol{W}^\top\boldsymbol{W}\boldsymbol{\Sigma}\boldsymbol{\beta}, \\
r_4(t) &= \boldsymbol{\beta}^\top\boldsymbol{\Sigma}\boldsymbol{W}^\top\boldsymbol{a}, \\
r_5(t) &= \boldsymbol{\beta}^\top\boldsymbol{\Sigma}\boldsymbol{W}^\top\boldsymbol{W}\boldsymbol{\Sigma}\boldsymbol{\beta}.
\end{aligned} \tag{45}$$

Introduce the constants $a = \boldsymbol{\beta}^\top\boldsymbol{\Sigma}\boldsymbol{\beta}$, $b = \boldsymbol{\beta}^\top\boldsymbol{\Sigma}^2\boldsymbol{\beta}$. Using the weight dynamics, it is straightforward to show that

$$\dot{\boldsymbol{r}}(t) \sim s \begin{bmatrix} 0 & a & 1 & 0 & 0 \\ 0 & 0 & 0 & 2 & 0 \\ b & 0 & 0 & a & 0 \\ 0 & b & 0 & 0 & 1 \\ 0 & 0 & 0 & 2b & 0 \end{bmatrix} \boldsymbol{r}(t) , \ t \ll \log(\sigma^{-2}). \tag{46}$$

This matrix has eigenvalues $\lambda \in \{0, -\sqrt{b}, \sqrt{b}, -2\sqrt{b}, 2\sqrt{b}\}$. Since there are only two positive eigenvalues $\sqrt{b}, 2\sqrt{b}$, it suffices to consider evolution along those two eigendirections, where the kernel and neural network function will be amplified. Evolution along these direcions give

$$\boldsymbol{r}(t) \sim c_1 e^{s\sqrt{b}t}\begin{bmatrix} 1 \\ 0 \\ \sqrt{b} \\ 0 \\ 0 \end{bmatrix} + c_2 e^{2s\sqrt{b}t}\begin{bmatrix} a \\ \sqrt{b} \\ a\sqrt{b} \\ b \\ b\sqrt{b} \end{bmatrix}, \tag{47}$$

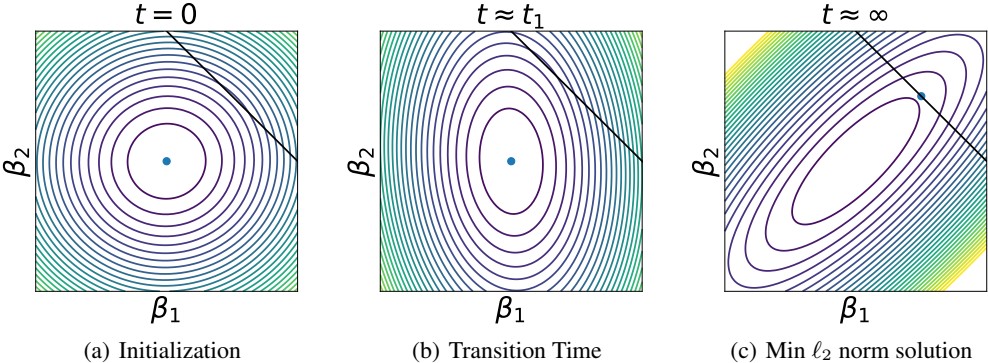

(a) Initialization      (b) Transition Time      (c) Min $\ell_2$ norm solution

Figure C.1: Kernel evolution on anisotropic data consists of two alignment phases. (a) At initialization the level curves of $\boldsymbol{\beta}^\top \boldsymbol{M}^{-1} \boldsymbol{\beta}$ exhibit spherical symmetry. (b) After the initial phase I alignment, the matrix $\boldsymbol{M}$ exhibits a spike in the $\boldsymbol{\Sigma}\boldsymbol{\beta}$ direction. (c) At long times, the network function and kernel's spiked direction need to converge to the minimum $\ell_2$ norm solution as we explain in G.1. This requires realignment of the kernel at late times, eliminating the preconditions for the silent alignment effect.

where $c_1, c_2$ are constants determined by intialization. At large time, the large eigenvalue mode $\lambda = 2\sqrt{b}$ will dominate. Decomposing $\boldsymbol{W} = \boldsymbol{W}_0 + \boldsymbol{a}(t) \left[ v_1(t)\boldsymbol{\beta} + v_2(t)\boldsymbol{\Sigma}\boldsymbol{\beta} \right]^\top$ we find that the only self consistent solution is $v_1(t) = 0, v_2(t) = b^{-1/2}$. This implies that the kernel evolution will take the form

$$K(\boldsymbol{x}, \boldsymbol{x}', t) \sim K(\boldsymbol{x}, \boldsymbol{x}', 0) + |\boldsymbol{a}(t)|^2 \boldsymbol{x}^\top \boldsymbol{M} \boldsymbol{x}',$$

$$\boldsymbol{M} = \left[ \frac{1}{\sqrt{\boldsymbol{\beta}^\top \boldsymbol{\Sigma}^2 \boldsymbol{\beta}}} \boldsymbol{\Sigma}\boldsymbol{\beta}\boldsymbol{\beta}^\top \boldsymbol{\Sigma} + \boldsymbol{I} \right]. \tag{48}$$

We see that the kernel evolves along the directions $\boldsymbol{\Sigma}\boldsymbol{\beta}$ early in training for unwhitened data. We visualize the two stages of learning for unwhitened data in Figure C.1.

## C.4 PHASE II: WHITENED DATA

Consider a two layer network $f = \boldsymbol{a}^\top \boldsymbol{W}\boldsymbol{x}$ where balance has been achieved $\boldsymbol{W} = u(t)\hat{\boldsymbol{a}}\boldsymbol{\beta}^\top$ and $\boldsymbol{a}(t) = u(t)\hat{\boldsymbol{a}}$. Once this balance condition is stable for fixed $\hat{\boldsymbol{a}}$, we can calculate the time derivative of $u(t)$

$$\frac{d}{dt}\boldsymbol{a}(t) = \dot{u}(t)\hat{\boldsymbol{a}} = u(t) \left[ s - u(t)^2 \right] \hat{\boldsymbol{a}}. \tag{49}$$

Letting $c(t) = u(t)^2$, we find that $\dot{c}(t) = 2u(t)^2 \left[ s - u(t)^2 \right] = 2c(t) \left[ s - c(t) \right]$, which is the two layer dynamics derived in Saxe et al. (2014). This dynamics has solution $c(t) = \frac{se^{2st}}{e^{2st} - 1 + s/c_0}$.

## C.5 SOLUTIONS TO THE FULL TRAINING DYNAMICS OF LINEAR NETWORKS AT SMALL INITIALIZATION

By combining the analyses of the subsection C.1 with the exact solutions discussed in C.4 we can match both solutions to obtain formulas for $r(t)$ and $q(t)$ for the entire network's training path that are exact up to $O(\sigma^2)$ corrections. Up to $O(\sigma^2)$ we then have that

$$r(t) = s \frac{2\sinh(2st)}{(e^{2st} - 1) + 2s/q_0}, \tag{50}$$

$$q(t) = s \frac{2\cosh(2st)}{(e^{2st} - 1) + 2s/q_0}. \tag{51}$$

This yields that the initialization constant $q_0/2$ plays the effective role of $c_0$ in the Ganguli-Saxe solution for phase II. Equation 34 yields the expected value of this initialization constant. We have

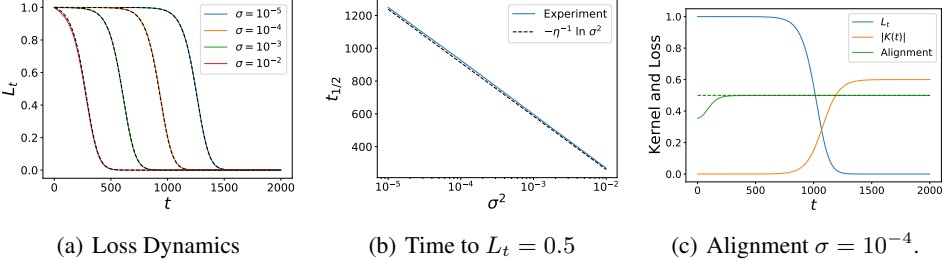

(a) Loss Dynamics      (b) Time to $L_t = 0.5$      (c) Alignment $\sigma = 10^{-4}$.

Figure C.2: Initialization scale controls the time spent in phase I, where the network escapes the saddle point near $\boldsymbol{W}, \boldsymbol{a} = \boldsymbol{0}$ and the kernel aligns to the task. (a) The loss curves for two-layer linear networks with small initialization follow sigmoidal trajectories as in Saxe et al. (2014) which transition from their maximum to minimum at a time which decreases with initialization scale. Theory is shown in black dashed lines. (b) Verification of the Phase I time $t_{1/2}$, measured as the time for the loss to reach one half its original value. This scales logarithmically with $\sigma^2$. (c) The alignment of the kernel eigenfunctions happens before the loss appreciably decreases for $\sigma = 10^{-4}$, evidenced by the kernel alignment curve. The analytically obtained maximum alignment value is overlayed in dashed green.

empirically verified that these exact equations hold to high accuracy across a variety network sizes, initialization scales, and whitened datasets. We illustrate some of these in figure C.3.

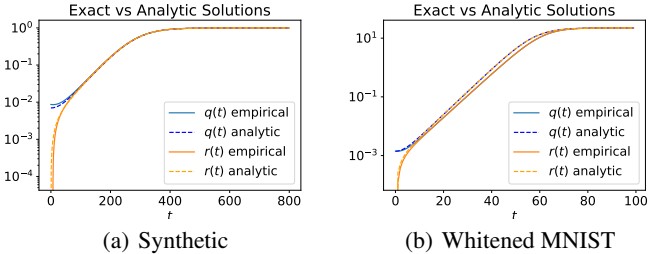

(a) Synthetic            (b) Whitened MNIST

Figure C.3: Overlay of empirical and exact solutions for $q(t), r(t)$ in two layer linear feedforward networks for synthetic and two-class MNIST whitened datasets. (a) We take $D = 25, N = 10$. (b) We take $D = 784, N = 100$.

## D   BALANCING OF WEIGHTS IN DEEP LINEAR NETWORKS

The balance condition discussed in the main text holds for deep linear trained with any loss function of the form $\mathcal{L} = \sum_\mu \ell(f^\mu, y^\mu)$ (not just MSE) since, as was shown by Arora et al. (2018); Du et al. (2018)

$$
\begin{aligned}
\frac{1}{\eta} \frac{d}{dt} \left( \boldsymbol{W}^\ell \boldsymbol{W}^{\ell\top} \right) &= \sum_\mu \frac{\partial \ell}{\partial f^\mu} \left[ \frac{\partial f^\mu}{\partial \boldsymbol{W}^\ell} \boldsymbol{W}^{\ell\top} + \boldsymbol{W}^\ell \frac{\partial f^\mu}{\partial \boldsymbol{W}^\ell}^\top \right] \\
&= \sum_\mu \frac{\partial \ell}{\partial f^\mu} \left[ \frac{\partial f^\mu}{\partial \boldsymbol{W}^{\ell+1}}^\top \boldsymbol{W}^{\ell+1} + \boldsymbol{W}^{\ell+1\top} \frac{\partial f^\mu}{\partial \boldsymbol{W}^{\ell+1}} \right] \\
&= \frac{1}{\eta} \frac{d}{dt} \left( \boldsymbol{W}^{\ell+1\top} \boldsymbol{W}^{\ell+1} \right).
\end{aligned}
\tag{52}
$$

The second line follows from the first since the following quantities are identical

$$\frac{\partial f^{\mu}}{\partial \boldsymbol{W}^{\ell}} \boldsymbol{W}^{\ell\top} = \boldsymbol{W}^{\ell+1\top}...\boldsymbol{W}^{L-1\top} \boldsymbol{w}^L \left(\boldsymbol{x}^{\mu\top} \boldsymbol{W}^{1\top}...\boldsymbol{W}^{\ell-1\top}\right) \boldsymbol{W}^{\ell\top}, \tag{53}$$

$$\boldsymbol{W}^{\ell+1\top} \frac{\partial f^{\mu}}{\partial \boldsymbol{W}^{\ell+1}} = \boldsymbol{W}^{\ell+1\top} \left(\boldsymbol{W}^{\ell\top}...\boldsymbol{w}^L\right) \boldsymbol{x}^{\mu\top} \boldsymbol{W}^{1\top}...\boldsymbol{W}^{\ell\top}. \tag{54}$$

By inspection these two quantities are equal. Thus we have, for any loss function, a deep linear network has the following conservation laws

$$\frac{d}{dt} \left[\boldsymbol{W}^{\ell} \boldsymbol{W}^{\ell\top} - \boldsymbol{W}^{\ell+1\top} \boldsymbol{W}^{\ell+1}\right] = 0. \tag{55}$$

We show in the next section that this balancing condition is very helpful in identifying the time evolution of the neural tangent kernel. In the case where the network has a single output, we can inductively prove that each layer's weight matrix is $\boldsymbol{W}^{\ell} = u(t)\boldsymbol{r}_{\ell+1}(t)\boldsymbol{r}_{\ell}(t)^{\top}$. We will assume this formula is true for layer $\ell + 1$ and prove it must hold for layer $\ell$ since

$$\boldsymbol{W}^{\ell} \boldsymbol{W}^{\ell\top} = \boldsymbol{W}^{\ell+1\top} \boldsymbol{W}^{\ell+1} = u(t)^2 \boldsymbol{r}_{\ell+1}(t)\boldsymbol{r}_{\ell+1}(t)^{\top}. \tag{56}$$

This implies that $\boldsymbol{W}^{\ell}$ is rank one with right singular vector equal to $\boldsymbol{r}_{\ell+1}(t)$. Thus, the decomposition for each layer the form $\boldsymbol{W}^{\ell} = u(t)\boldsymbol{r}_{\ell+1}(t)\boldsymbol{r}_{\ell}(t)^{\top}$ for some unit vector $\boldsymbol{r}_{\ell}(t)$. Similar analysis can be performed for the multi-class setting.

## E   NTK FORMULA FOR DEEP LINEAR NETWORKS

The neural tangent kernel for a linear network $f(\boldsymbol{x}) = \boldsymbol{w}^{L\top} \boldsymbol{W}^{L-1}...\boldsymbol{W}^1 \boldsymbol{x}$ is defined as an inner-product over all gradients

$$\begin{aligned}
K(\boldsymbol{x}, \boldsymbol{x}') &= \sum_{\ell=1}^{L} \left\langle \frac{\partial f(\boldsymbol{x})}{\partial \boldsymbol{W}^{\ell}}, \frac{\partial f(\boldsymbol{x}')}{\partial \boldsymbol{W}^{\ell}} \right\rangle \\
&= \left|\prod_{\ell=2}^{L} \boldsymbol{W}^{\ell}\right|^2 \boldsymbol{x} \cdot \boldsymbol{x}' + \sum_{\ell=2}^{L} \left|\prod_{\ell'>\ell}^{L} \boldsymbol{W}^{\ell'}\right|^2 \boldsymbol{x}^{\top} \boldsymbol{W}^{1\top}...\boldsymbol{W}^{\ell-1\top} \boldsymbol{W}^{\ell-1}...\boldsymbol{W}^1 \boldsymbol{x}'.
\end{aligned} \tag{57}$$

Under the balanced assumption that $\boldsymbol{W}^{\ell} = u(t)\boldsymbol{r}_{\ell+1}(t)\boldsymbol{r}_{\ell}(t)^{\top} + O(\sigma)$, expanding the kernel to leading order in $\sigma$ yields the following form:

$$K(\boldsymbol{x}, \boldsymbol{x}') = u(t)^{2L-2} \boldsymbol{x}^{\top} \left[(L-1)\boldsymbol{I} + \boldsymbol{r}_1(t)\boldsymbol{r}_1(t)^{\top}\right] \boldsymbol{x}' + O(\sigma), \tag{58}$$

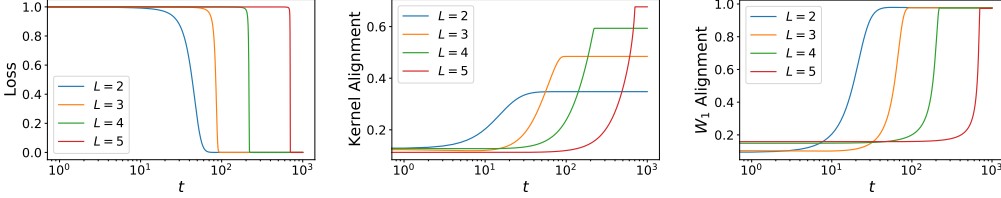

Figure E.1: The depth dependence of loss dynamics, kernel alignment and the alignment of first layer weights in a linear network on synthetic whitened data in $D = 30$ dimensions. (a) The loss reaches half its initial value after $t \sim \sigma^{-L+2}$ steps for $L > 2$. The decay rate of the loss becomes sharper with depth. (b) Final kernel alignment increases monotonically with depth $L$ and approaches 1 as $L \to \infty$. (c) The alignment of the first layer weights $\boldsymbol{W}_1$ with the optimal direction $\boldsymbol{\beta}$ approaches 1 for all models.

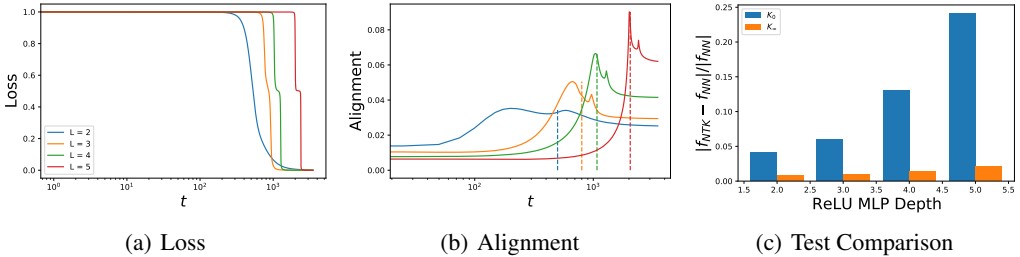

| (a) Loss | (b) Alignment | (c) Test Comparison |

Figure E.2: ReLU networks across depths trained on two-class whitened CIFAR. The hidden widths are all of size 100. We use 3k train points and 7k test points. (a) The loss exhibits the same scaling as $\sigma^{-L+2}$ as in the linear setting. (b) Deep networks with nonlinearities are seen to undergo the silent alignment effect early on in training. The dashed lines indicate when the kernel has grown to 10% of its final value. (c) The trained network outputs on test data match closely the kernel regression with the final learned kernel, but do not match regression the initial kernel.

## E.1 DEEP LINEAR NETWORK DYNAMICS UNDER BALANCE

In this section, we will consider the dynamics of the variable $u$ once the balance condition is satisfied. Let $\boldsymbol{w}_L = \hat{\boldsymbol{w}} u(t)$. Then the dynamics for $u(t)$ under the balancing assumption is

$$\frac{d}{dt}\boldsymbol{w}_L = \hat{\boldsymbol{w}}\frac{d}{dt}u(t) = \left(\prod_{\ell=1}^{L-1}\boldsymbol{W}^\ell\right)(s - u(t)^L)\prod_{\ell=1}^{L-1}\boldsymbol{W}^\ell\boldsymbol{\beta} = u(t)^{2L-2}(s - u(t)^L)\hat{\boldsymbol{w}}, \quad (59)$$

which implies the fact $\dot{u}(t) = u(t)^{L-1}(s - u(t)^L)$. Changing variables to $c(t) = u(t)^L$ we obtain

$$\dot{c}(t) = u(t)^{L-1}\dot{u} = u(t)^{2L-2}(s - u(t)^L) = c(t)^{2-2/L}(s - c(t)). \quad (60)$$

When $c_0^L$ is initialized to a very small value compared to $s$ we can

$$\frac{d}{dt}c(t) \sim c(t)^{2-2/L}s \implies \left[c(t)^{-1+2/L} - c_0^{-1+2/L}\right] = -\frac{L-2}{L}st$$

$$\implies c(t) = \left[c_0^{-\frac{L-2}{L}} - \frac{(L-2)}{L}st\right]^{-\frac{L}{L-2}}. \quad (61)$$

This implies a timescale to learn of $t \sim s^{-1}\frac{L}{L-2}\sigma^{-L+2}$.

We can approximate the timescale for the first layer's singular vector $\boldsymbol{r}_1(t)$ to align to $\boldsymbol{\beta}$ as well. Let $\boldsymbol{v}$ be a vector orthogonal to $\boldsymbol{\beta}$.

$$\frac{d}{dt}\frac{|\boldsymbol{W}^{(1)}\boldsymbol{\beta}|^2}{|\boldsymbol{W}^{(1)}\boldsymbol{v}|^2} = \frac{2\boldsymbol{\beta}^\top\boldsymbol{W}^{(1)\top}...\boldsymbol{w}_L}{|\boldsymbol{W}^{(1)}\boldsymbol{v}|^2} \sim O(\sigma^{L-2}). \quad (62)$$

This suggests that alignment in a deep network should also occur on a timescale of $t \sim \sigma^{-L+2}$. While there is no strict separation of timescales in terms of the scaling of alignment and learning with $\sigma$, we find that alignment tends to precede a significant drop in the loss as we show in Figure 3.

## E.2 FINAL NTK FOR DEEP LINEAR NETWORKS

Independent of the structure of the data, the first vector $\boldsymbol{r}_1 \to \boldsymbol{\beta}$ and the final NTK has the following form:

$$K(\boldsymbol{x}, \boldsymbol{x}') = s^{2L-2}\boldsymbol{x}^\top\left[(L-1)\boldsymbol{\beta}\boldsymbol{\beta}^\top + \boldsymbol{I}\right]\boldsymbol{x}' + O(\sigma). \quad (63)$$

This formula is merely a consequence of the balancing condition and convergence to the optimum $u(t)^L \to s$. We provide empirical support that final kernel alignment increases with depth in Figure E.3.

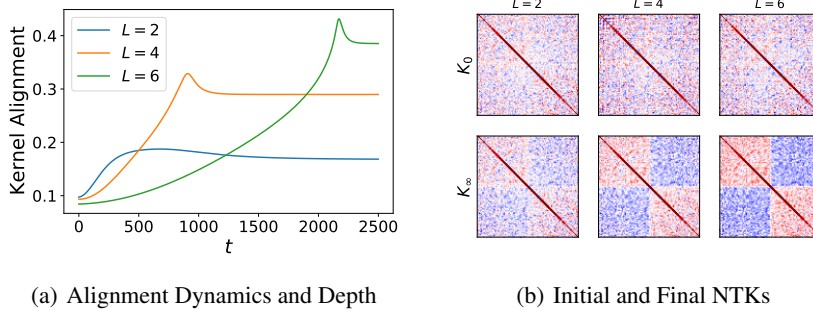

(a) Alignment Dynamics and Depth          (b) Initial and Final NTKs

Figure E.3: The final NTK for a deep linear network aligns with the class specific directions with strength that depends on network depth $L$. This experiment is a partially whitened ($\gamma = 0.25$ see Section 5) subset of 100 CIFAR-10 images. (a) The dynamics of alignment for different depth models. (b) The final gram matrix has the form $\boldsymbol{K}_\infty \propto (L-1)\boldsymbol{y}\boldsymbol{y}^\top + \boldsymbol{K}_0$, illustrating why alignment of the final NTK to class structure increases with depth $L$.

### E.3    Formula for the Intermediate NNGP Kernels

Let $\boldsymbol{\beta}$ represent the unit vector pointing in the optimal direction for the scalar output case. To calculate this, all that needs to be assumed is that the network has converged to its optimum and that the weights satisfy the balance property so that $\boldsymbol{W}^\ell = u^* \boldsymbol{r}_{\ell+1}^* \boldsymbol{r}_\ell^*$. By the convergence assumption $u^* = s^{1/L}$ and $\boldsymbol{r}_1^* = \boldsymbol{\beta}$. We stress that this holds for arbitrary input correlation structure $\boldsymbol{\Sigma}$, the final NNGP kernel for layer $\ell$ can also be computed. Expanding the weights and collecting terms at leading order in $\sigma$ yields:

$$K_\ell(\boldsymbol{x}, \boldsymbol{x}') \sim s^{2\ell/L} \boldsymbol{x}^\top \boldsymbol{\beta}\boldsymbol{\beta}^\top \boldsymbol{x}' + O(\sigma). \tag{64}$$

Evaluating on the training set gives $\boldsymbol{K}_\ell = \left(\boldsymbol{y}\boldsymbol{y}^\top\right)^{\ell/L} \in \mathbb{R}^{P \times P}$.

## F    Generalization Error in Transfer Learning Task

The structure of the final kernel can alter the ability of the network to flexibly transfer to new tasks with a small amount of data. In this section, we examine how learned intermediate representations compares with the inductive bias of the original isotropic kernel $\boldsymbol{x} \cdot \boldsymbol{x}'$. In particular, we study the offline generalization performance of kernels of the form

$$K(\boldsymbol{x}, \boldsymbol{x}'; A) = \boldsymbol{x}^\top \left[A\boldsymbol{\beta}\boldsymbol{\beta}^\top + \boldsymbol{I}\right] \boldsymbol{x}'. \tag{65}$$

In Section 4 we showed that $A$ could be altered by changing the network depth. Concretely, our transfer learning problem consists of training a linear probe on one of the intermediate layers of the network (Alain & Bengio (2016); Cohen et al. (2020)). This would also produce a kernel regression solution for kernel $K(\boldsymbol{x}, \boldsymbol{x}', A)$ with $A$ which depends on the chosen layer and the depth of the network. For simplicity, we assume that the data are generated according to a simple Gaussian distribution $\boldsymbol{x} \sim \mathcal{N}(0, \boldsymbol{I})$ and that the target values are generated with a linear function $y(\boldsymbol{x}) = \boldsymbol{w} \cdot \boldsymbol{x}$. We decompose the new task vector $\boldsymbol{w} = \alpha\boldsymbol{\beta} + \sqrt{(1 - \alpha^2)}\boldsymbol{w}_\perp$ where $\boldsymbol{w}_\perp \cdot \boldsymbol{\beta} = 0$. The expected generalization error after training with $P$ samples can be computed with methods from the physics of disordered systems (Bordelon et al., 2020; Canatar et al., 2021; Loureiro et al., 2021). For any $A > 0$, the easiest transfer task is $\boldsymbol{w} = \boldsymbol{\beta}$ ($\alpha = 1$). If $\boldsymbol{w} = \boldsymbol{\beta}$, increasing the alignment $A$ strictly decreases the generalization error. This is illustrated in Figure F.1.

### F.1    Derivation of Learning Curves

We will discuss the average case generalization error in the transfer learning setting. Prior work has shown that the generalization performance of kernel regression can be calculated through a kernel

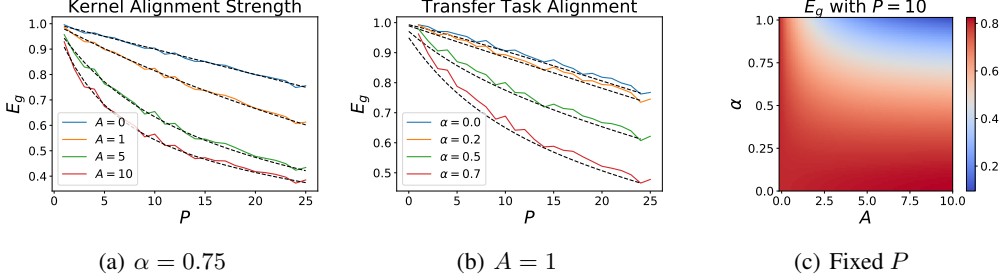

(a) $\alpha = 0.75$        (b) $A = 1$        (c) Fixed $P$

Figure F.1: The offline generalization error in a transfer task with the learned linear kernel $K = \boldsymbol{x}^\top \left[ A\boldsymbol{\beta}^\top \boldsymbol{\beta}^\top + \boldsymbol{I} \right] \boldsymbol{x}'$ and $y_{new} = \left[ \alpha\boldsymbol{\beta} + \sqrt{(1 - \alpha^2)} \boldsymbol{w}_\perp \right] \cdot \boldsymbol{x}$. (a) For a new transfer task which is correlated with the learned function $\beta$, neural networks with large feature learning $A$ give lower generalization error at small sample sizes. (b) For fixed $A = 1$, the tasks $\boldsymbol{w}$ which are strongly correlated with $\boldsymbol{\beta}$ are easier to learn during transfer. (c) The lowest generalization error in a transfer learning setup occurs when feature learning strength $A$ and correlation between tasks $\alpha$ are both large.

eigenvalue problem (Bordelon et al., 2020; Canatar et al., 2021; Loureiro et al., 2021)

$$\langle K(\boldsymbol{x}, \boldsymbol{x}')\phi_k(\boldsymbol{x})\rangle_{\boldsymbol{x}} = \lambda_k \phi_k(\boldsymbol{x}). \tag{66}$$

Once this integral eigenvalue problem is solved for eigenvalues $\lambda_k$ and orthonormal eigenfunctions $\phi_k$, the average case generalization error at $P$ training examples is

$$E_g = \frac{1}{1 - \gamma} \sum_k \frac{\langle y(\boldsymbol{x})\phi_k(\boldsymbol{x})\rangle^2}{(\lambda_k P + \kappa)^2} \;,\; \kappa = \lambda + \kappa \sum_k \frac{\lambda_k}{\lambda_k P + \kappa} \;,\; \gamma = P \sum_k \frac{\lambda_k^2}{(\lambda_k P + \kappa)^2}. \tag{67}$$

In our case, we are interested in the generalization performance of the linear kernel

$$K(\boldsymbol{x}, \boldsymbol{x}') = \boldsymbol{x}^\top \left[ \boldsymbol{\beta}\boldsymbol{\beta}^\top + \boldsymbol{I} \right] \boldsymbol{x}'. \tag{68}$$

Since $K$ is a linear kernel, its eigenfunctions should be linear functions $\phi_k(\boldsymbol{x}) = \boldsymbol{\phi}_k \cdot \boldsymbol{x}$. Assuming that the data distribution has identity covariance, we find

$$\left\langle \boldsymbol{x}^\top \left[ A\boldsymbol{\beta}\boldsymbol{\beta}^\top + \boldsymbol{I} \right] \boldsymbol{x}' \boldsymbol{\phi}_k^\top \boldsymbol{x} \right\rangle = \boldsymbol{\phi}_k^\top \left[ A\boldsymbol{\beta}\boldsymbol{\beta}^\top + \boldsymbol{I} \right] \boldsymbol{x}' = \lambda_k \boldsymbol{\phi}_k^\top \boldsymbol{x}'. \tag{69}$$

This implies that the $\boldsymbol{\phi}_k$ vectors are eigenvectors of $\boldsymbol{M}$. The first eigenvector is $\boldsymbol{\phi}_1 = \boldsymbol{\beta}$ with eigenvalue $\lambda_1 = A + 1$. The other $D - 1$ eigenvectors can be chosen as any frame in the $D - 1$ dimensional subspace orthogonal to $\boldsymbol{\beta}$. Each of these $D - 1$ eigenvectors has eigenvalue $\lambda_k = 1$. Using these results, and the fact that $\boldsymbol{w} = \alpha\boldsymbol{\phi}_1 + \sqrt{1 - \alpha^2}\boldsymbol{w}_\perp$, we can calculate the expected generalization error.

$$E_g = \frac{1}{1 - \gamma} \left[ \frac{\alpha^2}{((1 + A)P + \kappa)^2} + \frac{1 - \alpha^2}{(P + \kappa)^2} \right]$$
$$\kappa = \lambda + \kappa \left[ \frac{1 + A}{(1 + A)P + \kappa} + \frac{D - 1}{P + \kappa} \right] \;,\; \gamma = P \left[ \frac{(1 + A)^2}{((1 + A)P + \kappa)^2} + \frac{D - 1}{(P + \kappa)^2} \right].$$

By the result proven in Canatar et al. (2021), the lowest possible error for fixed $A$ occurs by maximizing the fraction of variance along the large eigenvalue direction, corresponding to $\alpha = \pm 1$.

## G    LINEAR NTKS DURING GD LEARN THE SAME FUNCTION

In the overparameterized setting where $D > P$, all linear networks discussed in the Section 4 converge to the minimum norm interpolator when the data is whitened. Specifically, letting the learned neural network function be written as $f(x) = \hat{\boldsymbol{\beta}}^\top \boldsymbol{x}$, and the data matrix $\boldsymbol{X} \in \mathbb{R}^{D \times P}$ and

target labels $\boldsymbol{y} \in \mathbb{R}^P$ represent the training data. The solution vector $\boldsymbol{\beta}$ solves the constrained optimization problem

$$\min_{\hat{\boldsymbol{\beta}}} ||\hat{\boldsymbol{\beta}}||^2 \ , \ \text{s.t.} \ \boldsymbol{X}^\top \hat{\boldsymbol{\beta}} = \boldsymbol{y}, \tag{70}$$

which is the kernel regression solution for the initial kernel $K_0(x, x') = \boldsymbol{x} \cdot \boldsymbol{x}'$. This is unsurprising due to a symmetry argument: when $\boldsymbol{\beta}_0 \approx 0$, the only privileged point on the affine space $\boldsymbol{X}^\top \boldsymbol{\beta} = \boldsymbol{y}$ is the point closest to the origin, which is precisely the solution above. Surprisingly, the final pseudo-inverse solution $s\boldsymbol{\beta}$ also minimizes the RKHS norms for any of the kernels throughout gradient descent. Up to an overall scale, the kernels throughout evolution take the form $K(\boldsymbol{x}, \boldsymbol{x}'; t) = \boldsymbol{x}^\top \left[ A(t)\boldsymbol{\beta}\boldsymbol{\beta}^\top + \boldsymbol{I} \right] \boldsymbol{x}'$ (see Section 4.1) which would induce the following kernel interpolation problems

$$\min_{\hat{\boldsymbol{\beta}}} \hat{\boldsymbol{\beta}}^\top \left[ A(t)\boldsymbol{\beta}\boldsymbol{\beta}^\top + \boldsymbol{I} \right]^{-1} \hat{\boldsymbol{\beta}} \ , \ \text{s.t.} \ \boldsymbol{X}^\top \hat{\boldsymbol{\beta}} = \boldsymbol{y}. \tag{71}$$

The solution to this optimization problem is indeed the kernel regression solution with kernel $K(t)$ since the learned function takes the form $f(x) = \sum_\mu \alpha^\mu K(\boldsymbol{x}, \boldsymbol{x}^\mu, t)$ with $\boldsymbol{\alpha} = \boldsymbol{K}(t)^{-1}\boldsymbol{y}$. Using the Sherman-Morrison rule, we show that the solution to each of these problems $t \geq 0$ gives the same result, namely the pseudo-inverse solution. This can be seen from the following

$$\hat{\boldsymbol{\beta}}^\top \left[ A(t)\boldsymbol{\beta}\boldsymbol{\beta}^\top + \boldsymbol{I} \right]^{-1} \hat{\boldsymbol{\beta}} = |\hat{\boldsymbol{\beta}}|^2 - \frac{A(t)}{1 + A(t)} \left( \boldsymbol{\beta} \cdot \hat{\boldsymbol{\beta}} \right)^2. \tag{72}$$

Now, we let $\hat{\boldsymbol{\beta}} = s\boldsymbol{\beta} + \boldsymbol{\beta}_\perp$, where $\boldsymbol{\beta} \cdot \boldsymbol{\beta}_\perp = 0$. This is the general decomposition for the set of interpolators which have the property $\boldsymbol{X}^\top [\boldsymbol{\beta} + \boldsymbol{\beta}_\perp] = \boldsymbol{y}$.

$$\min_{\boldsymbol{\beta}_\perp} |\boldsymbol{\beta}_\perp|^2. \tag{73}$$

The solution is merely to set $\boldsymbol{\beta}_\perp = 0$. Thus the optimal solution is therefore the same for any finite value of $A$. However, the final RKHS norm of the learned function $\boldsymbol{\beta}^\top \left[ A(t)\boldsymbol{\beta}\boldsymbol{\beta}^\top + \boldsymbol{I} \right]^{-1} \boldsymbol{\beta}$ decreases with time, indicating that the kernel becomes more aligned with the pseudo-inverse direction as $A$ increases.

### G.1 DEEP LINEAR NETWORKS FROM SMALL INITIALIZATION LEARN PSEUDO-INVERSE

In this subsection of the appendix, we will use our theoretical technology for balanced linear networks to demonstrate the universal learned function for *any data*, not just whitened input, providing an alternative derivation to the result proven in Theorem 7 of Yun et al. (2020). This analysis is performed in the $\sigma \to 0$ limit, where $\boldsymbol{W}_\ell = u(t)\boldsymbol{r}_{\ell+1}(t)\boldsymbol{r}_\ell(t)^\top$ as we showed in Section D. Under this condition the learned function $f(\boldsymbol{x}) = \hat{\boldsymbol{\beta}} \cdot \boldsymbol{x}$ is defined through weights $\hat{\boldsymbol{\beta}} = \boldsymbol{W}_1^\top \boldsymbol{W}_2^\top ... \boldsymbol{w}_L = u(t)^L \boldsymbol{r}_1(t)$. We see that the direction of the learned function is controlled entirely by $\boldsymbol{r}_1(t)$. It suffices to prove that $\boldsymbol{r}_1(t) \in \text{span}\{\boldsymbol{x}_1, ..., \boldsymbol{x}_P\}$ for all $t$ to show that the network learns the pseudo-inverse solution $\boldsymbol{\beta}^* = \boldsymbol{X}(\boldsymbol{X}^\top \boldsymbol{X})^{-1}\boldsymbol{y}$, where $\boldsymbol{X} \in \mathbb{R}^{D \times P}$ and $\boldsymbol{y} \in \mathbb{R}^P$ are the training data and targets respectively. Note that by gradient descent, we have

$$\frac{d}{dt}\boldsymbol{W}_1(t) = \boldsymbol{W}_2^\top ... \boldsymbol{w}_L \sum_\mu (y_\mu - f_\mu(t))\boldsymbol{x}_\mu^\top = u(t)^{L-1}\boldsymbol{r}_2(t) \sum_\mu (y_\mu - f_\mu(t))\boldsymbol{x}_\mu^\top. \tag{74}$$

From the balance condition $\boldsymbol{W}_1(t) = u(t)\boldsymbol{r}_2(t)\boldsymbol{r}_1(t)^\top$, we also have

$$\frac{d}{dt}\boldsymbol{W}_1(t) = \left[ \dot{u}(t)\boldsymbol{r}_2(t) + u(t)\dot{\boldsymbol{r}}_2(t) \right] \boldsymbol{r}_1(t)^\top + u(t)\boldsymbol{r}_2(t)\dot{\boldsymbol{r}}_1(t)^\top. \tag{75}$$

Equating the two above expressions for $\frac{d}{dt}\boldsymbol{W}_1(t)$ and taking an inner product with $\boldsymbol{r}_2(t)$ from the left gives the following

$$u(t)\dot{\boldsymbol{r}}_1(t) = u(t)^{L-1} \sum_\mu (y_\mu - f_\mu(t))\boldsymbol{x}^\mu - \left[ \dot{u}(t) + u(t)\boldsymbol{r}_2(t) \cdot \dot{\boldsymbol{r}}_2(t) \right] \boldsymbol{r}_1(t). \tag{76}$$

Thus, if $\boldsymbol{r}_1(t) \in \text{span}\{\boldsymbol{x}_1, ..., \boldsymbol{x}_P\}$ then $\dot{\boldsymbol{r}}_1(t) \in \text{span}\{\boldsymbol{x}_1, ..., \boldsymbol{x}_P\}$ so that the full dynamics of $\boldsymbol{r}_1(t)$ lie in the subspace spanned by the training data. At initialization, we have $\dot{\boldsymbol{W}}_1(t) \propto \boldsymbol{z}(t) \sum_\mu \boldsymbol{y}^\mu x_\mu^\top + O(\sigma^3)$ so the initial $\boldsymbol{r}_1$ vector will indeed align with the span of the training data in the $\sigma \to 0$ limit.

By the fact that $\hat{\boldsymbol{\beta}} = u(t)^L \boldsymbol{r}_1(t)$, the learned linear coefficients $\hat{\boldsymbol{\beta}}$ must also be in $\text{span}\{\boldsymbol{x}_1, ..., \boldsymbol{x}_P\}$ so $\hat{\boldsymbol{\beta}} = \sum_\mu \alpha_\mu \boldsymbol{x}_\mu$. These must also interpolate the data provided $D \geq P$, giving the following condition

$$\hat{\boldsymbol{\beta}} \cdot \boldsymbol{x}^\nu = \sum_\mu \boldsymbol{x}_\nu \cdot \boldsymbol{x}_\mu \alpha_\mu = y_\nu \implies \boldsymbol{\alpha} = (\boldsymbol{X}^\top \boldsymbol{X})^{-1} \boldsymbol{y} \implies \hat{\boldsymbol{\beta}} = \boldsymbol{X}(\boldsymbol{X}^\top \boldsymbol{X})^{-1} \boldsymbol{y}. \tag{77}$$

This is exactly the minimum $\ell_2$ norm interpolating solution which solves

$$\min_{\hat{\boldsymbol{\beta}}} |\hat{\boldsymbol{\beta}}|_2^2 \text{ , s.t. , } \boldsymbol{X}^\top \hat{\boldsymbol{\beta}} = \boldsymbol{y}. \tag{78}$$

While anisotropy of the data makes no impact on what function is ultimately learned in the linear network case, the anisotropy can have a signficant influence on whether the preconditions for silent alignment are satisfied in a nonlinear network, which can prevent the final function from being a NTK regressor with final NTK.

## H  FINAL KERNEL IN MULTI-CLASS NETWORKS

For a network with $C$ output channel, balancing and alignment guarantee that the configuration of the network is orthogonal and balanced as in the setting of Saxe et al. (2014). One can then integrate each mode separately to obtain the final kernel as

$$\boldsymbol{W}^\ell = \sum_{\alpha=1}^C u_\alpha(t) \boldsymbol{r}_{\ell+1}^\alpha(t) \boldsymbol{r}_\ell^\alpha(t)^\top \text{ , } K_{c,c'}(\boldsymbol{x}, \boldsymbol{x}') = \boldsymbol{x}^\top \boldsymbol{M}_{c,c'} \boldsymbol{x}',$$

$$M_{c,c'} = \delta_{c,c'} \sum_\alpha u_\alpha^{2(L-1)}(t) \boldsymbol{r}_1^\alpha \boldsymbol{r}_1^\alpha + \sum_{\ell=1}^{L-1} \sum_\alpha u_\alpha^{2(L-\ell)}(t) \boldsymbol{e}_c^\top \boldsymbol{r}_L^\alpha \boldsymbol{r}_L^{\alpha\top} \boldsymbol{e}_{c'} \sum_\beta u_\beta^{2(\ell-1)}(t) \boldsymbol{r}_1^\beta \boldsymbol{r}_1^{\beta\top}, \tag{79}$$

where the Cartesian unit vectors $\boldsymbol{e}_c \in \mathbb{R}^C$ are one-hot on class output $c$. This shows that the contributions to the kernel depend on how well the class output channels align with the unit vectors $\boldsymbol{r}_L^\alpha$. Further, the singular values $u_\alpha$ can evolve at different timescales depending on the structure of the data.

Specifically, for a depth $L$ network, both the the alignment time $t_\alpha^{(L)}$ and the time to learn a given singular value $s_\alpha$ scale as $s_\alpha^{-1}\sigma^{2-L}$, as shown in appendix E.1. The differences in alignment times $\Delta t_{\alpha\beta} := t_\alpha^{(L)} - t_\beta^{(L)}$ for modes $s_\alpha, s_\beta$ therefores scales as $\Delta t_{\alpha\beta}^{(L)} = \sigma^{2-L} \Delta t_{\alpha\beta}^{(2)}$.

### H.1  FINAL NNGP IN MULTI-OUTPUT CASE

We can also gain intuition about the learned representations in each layer by looking at the NNGP kernels, which merely take inner-products between layer activations for different inputs. Let $\boldsymbol{\beta} \in \mathbb{R}^{C \times D}$ represent the optimal weight matrix which has the property (in the over-parameterized case) $\boldsymbol{\beta}\boldsymbol{x}^\mu = \boldsymbol{y}^\mu$. At the optimum, the neural network must learn $\boldsymbol{\beta} = \boldsymbol{W}^L ... \boldsymbol{W}^1$. Computing the SVD of $\boldsymbol{\beta} = \sum_\alpha \beta_\alpha \boldsymbol{z}_\alpha \boldsymbol{v}_\alpha^\top$ reveals that $u_\alpha^* = \beta_\alpha^{1/L}$ and $\boldsymbol{r}_\alpha^1 = \boldsymbol{v}_\alpha$ and $\boldsymbol{r}_\alpha^L = \boldsymbol{z}_\alpha$. Using these facts, it is easy to derive the final NNGP kernel for layer $\ell$.

$$K_\ell(\boldsymbol{x}, \boldsymbol{x}') = \boldsymbol{x}^\top \boldsymbol{W}^{1\top} ... \boldsymbol{W}^{\ell\top} \boldsymbol{W}^\ell ... \boldsymbol{W}^1 \boldsymbol{x}'$$

$$= \boldsymbol{x}^\top \left[ \sum_\alpha (u_\alpha^*)^{2\ell} \boldsymbol{r}_\alpha^1 \boldsymbol{r}_\alpha^{1\top} \right] \boldsymbol{x}' = \boldsymbol{x}^\top \left[ \boldsymbol{\beta}^\top \boldsymbol{\beta} \right]^{\ell/L} \boldsymbol{x}'. \tag{80}$$

Evaluating on the training set $\boldsymbol{X} \in \mathbb{R}^{D \times P}$ gives $\boldsymbol{K}_\ell = \boldsymbol{X}^\top \left[ \boldsymbol{\beta}^\top \boldsymbol{\beta} \right]^{\ell/L} \boldsymbol{X}$, which interpolates between $\boldsymbol{X}^\top \boldsymbol{X}$ at layer $\ell = 0$ and $\boldsymbol{Y}^\top \boldsymbol{Y}$ at layer $L$.

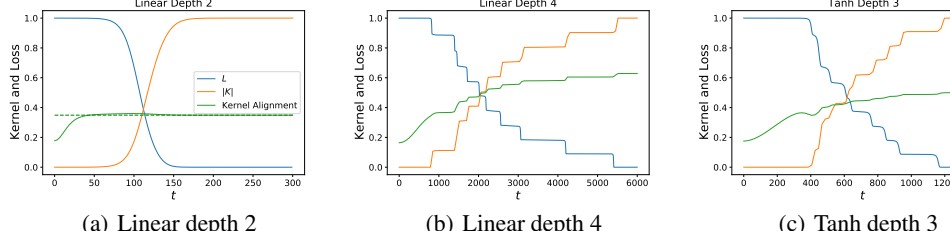

Figure H.1: Demonstration of a separation between alignment and spectral learning phases across networks trained on multi-class data. Here we train on whitened MNIST. Each network has $\sigma$ so that $\sigma^L = 10^{-4}$ for $L$ the depth of the network. (a) Depth two multi-class dynamics are very similar to single class. The analytically predicted final alignment is in dashed green. (b) For deeper multi-class networks, each singular values is learned far apart in time, and alignment does not as clearly precede loss decay. (c) Similar dynamics are obtained for tanh networks. Note for deeper networks there is a stronger separation of the times to learn each singular value, resulting in the ten separated drops in the loss.

## H.2  MORE REFINED BALANCING ANALYSIS

We can derive corrections to our decomposition of the kernel by including the initial conditions in our derived conservation laws. In particular, we will consider balancing for large width networks. Note that the network does not need to be in the lazy regime. The balancing condition is

$$\boldsymbol{W}^\ell(t)\boldsymbol{W}^{\ell\top}(t) - \boldsymbol{W}^{\ell+1\top}(t)\boldsymbol{W}^{\ell+1}(t) = \boldsymbol{W}_0^\ell\boldsymbol{W}_0^{\ell\top} - \boldsymbol{W}_0^{\ell+1\top}\boldsymbol{W}_0^{\ell+1}. \tag{81}$$

Note that $\boldsymbol{W}_0^\ell \in \mathbb{R}^{N_{\ell+1} \times N_\ell}$ has entries with zero mean and variance $\sigma^2/N_\ell$ in the standard parameterization. The products of initial matrices are therefore Wishart distributed. For sufficiently large widths, we can approximate the initial weight matrix products with their expectation over the random initialization

$$\boldsymbol{W}_0^\ell\boldsymbol{W}_0^{\ell\top} - \boldsymbol{W}_0^{\ell+1\top}\boldsymbol{W}_0^{\ell+1} \approx \sigma^2 \left(1 - \frac{N_{\ell+2}}{N_{\ell+1}}\right) \boldsymbol{I}. \tag{82}$$

This concentration becomes more accurate as the widths $N_\ell, N_{\ell+1} \to \infty$. For the last layer if the number of classes $C = N_L$ is sufficiently large, we also obtain similar concentration for the last layer $\boldsymbol{W}^{L\top}\boldsymbol{W}^L \approx \sigma^2 \frac{N_L}{N_{L-1}} \boldsymbol{I}$. Repeating the backward induction on the conservation law, we find the following recursively defined singular value decompositions

$$\boldsymbol{W}^\ell = \sum_\alpha u_\alpha^\ell(t) \boldsymbol{r}_{\ell+1}^\alpha(t) \boldsymbol{r}_\ell^\alpha(t)^\top,$$

$$u_\alpha^\ell(t)^2 = u_\alpha^{\ell+1}(t)^2 + \sigma^2 \left(1 - \frac{N_{\ell+2}}{N_{\ell+1}}\right) = u_\alpha^L(t)^2 + \sigma^2 g_\ell, \tag{83}$$

$$g_\ell = L - \ell - \sum_{k=\ell}^{L-1} \frac{N_{k+1}}{N_k}.$$

We note that the corrections $g_\ell$ vanish if all layers $k > \ell$ have the same width. Let $u_\alpha = u_\alpha^{(L)}(t)$. The condition for convergence is

$$\prod_\ell \left[u_\alpha^{(\ell)}\right]^2 = \prod_\ell \left[(u_\alpha)^2 + \sigma^2 g_\ell\right] = s_\alpha^2. \tag{84}$$

In the $\sigma^2 \to 0$ limit, we can solve that $u_\alpha \sim s^{1/L}$ as before. However, we can now obtain leading order corrections (in $\sigma^2$) which take the form of the form

$$[u_\alpha]^2 \sim s_\alpha^2 - \frac{g\sigma^2}{2Ls_\alpha^{1/L}} \; , \; \sigma^2 \to 0 \; , \; g = \sum_{\ell=1}^L g_\ell = \frac{L(L-1)}{2} - \sum_{\ell=1}^L \sum_{k=\ell}^L \frac{N_{k+1}}{N_k}, \tag{85}$$

which reveals that the size of the correction depends not only on $\sigma$ but also on the depth and network widths. Suppose all network widths were equal $N_\ell = N_k$, then the term $g = 0$ and there is no contribution from the first moment of the random weights.

## I  LAZINESS IN HOMOGENOUS NETWORKS

In this section we recapitulate the argument found in Chizat et al. (2019). The goal is to estimate how rapidly the gradient features on a test point $\nabla f(\boldsymbol{x})$ change compared to the loss early in training. Let $\boldsymbol{f} \in \mathbb{R}^P$ represent the function outputs on the training set. We will compute the time derivatives of the loss and the network gradients.

$$\left|\frac{d}{dt}\nabla_{\boldsymbol{\theta}}f(\boldsymbol{x})\right| = \left|\nabla^2 f(\boldsymbol{x}) \cdot \frac{d\boldsymbol{\theta}}{dt}\right| = \left|\nabla^2 f(\boldsymbol{x}) \cdot \sum_\mu (y^\mu - f^\mu)\nabla f^\mu\right|, \tag{86}$$

$$\left|\frac{d}{dt}\mathcal{L}\right| = \left|\frac{d\boldsymbol{\theta}}{dt}\right|^2 = |\nabla \boldsymbol{f} \cdot (\boldsymbol{y} - \boldsymbol{f})|^2. \tag{87}$$

Here, $|\cdot|_{op}$ denotes the operator norm of a matrix. We are interested in the ratio of the loss' time derivative to the gradient's time derivative. With an initialization scale of $\boldsymbol{f} \sim O(\sigma^L)$ we find

$$\frac{\left|\frac{d}{dt}\nabla f(\boldsymbol{x})\right|}{|\nabla \boldsymbol{f}|}\frac{\mathcal{L}}{\left|\frac{d}{dt}\mathcal{L}\right|} = \frac{|\nabla^2 f(\boldsymbol{x}) \cdot \sum_\mu (y^\mu - f^\mu)\nabla f^\mu||\boldsymbol{y} - \boldsymbol{f}|^2}{|\nabla \boldsymbol{f}||\nabla \boldsymbol{f} \cdot (\boldsymbol{y} - \boldsymbol{f})|^2} \approx \frac{|\nabla^2 f(\boldsymbol{x}) \cdot \sum_\mu y^\mu \nabla f^\mu||\boldsymbol{y}|^2}{|\nabla \boldsymbol{f}||\nabla \boldsymbol{f} \cdot \boldsymbol{y}|^2}, \tag{88}$$

where in the last step we approximated $y^\mu - f^\mu \approx y^\mu$ for small initialization scale since $y^\mu \sim O(1)$ and $f^\mu \sim O(\sigma^L)$. Now we will estimate the scale of each of the terms above. For a homogenous model $\nabla f \sim O(\sigma^{L-1})$ and $\nabla^2 f \sim O(\sigma^{L-2})$. Counting powers of $\sigma$ in numerator and denominator, we find that this quantity of interest scales as

$$\frac{\left|\frac{d}{dt}\nabla f(\boldsymbol{x})\right|}{|\nabla \boldsymbol{f}|}\frac{\mathcal{L}}{\left|\frac{d}{dt}\mathcal{L}\right|} = O(\sigma^{-L}). \tag{89}$$

This result indicates that, from small initialization, the gradient NTK features and thus the kernel itself will evolve much more rapidly than the loss. This effect can be amplified by increasing depth and decreasing initialization scale.

## J  RESNET EXPERIMENTAL DETAILS

Below, we provide the alignment and loss dynamics for wide resnet for CIFAR-10 with 100 training points. Because the loss decreases significantly before the kernel reaches its final alignment value, the final NTK is not perfectly correlated with the final neural network function. The wide ResNet model is taken from Novak et al. (2020) and is based on the original architecture of Zagoruyko & Komodakis (2017) with a widening factor of $k = 4$ and a single block per ResNet group $b = 1$, giving a final network with 8 trainable layers. For both Figure 1 (d) and (e) as well as Figure J.1 use Adam with a learning rate of $\eta = 10^{-5}$ and initial weight scale of $\sigma = 0.3$ in standard parameterization for all intermediate blocks. For the first conv layer, we used $\sigma = 6.0$. We find that small initial weight variance in the first layer gives rise to less stable learning and worse kernel alignment.

Below, in Figure J.2, we provide comprehensive results for different depths which we control by increasing the number of blocks per group $b$, corresponding to WRNs with $6b + 1$ trainable conv layers.

### J.1  ADAPTIVE OPTIMIZERS AND THE RELEVANT KERNEL

Many adaptive gradient methods compute updates to parameters $\theta_j$ according to

$$\dot{\theta}_j(t) = -\eta_j(t)\frac{\partial \mathcal{L}}{\partial \theta_{j,t}} \tag{90}$$

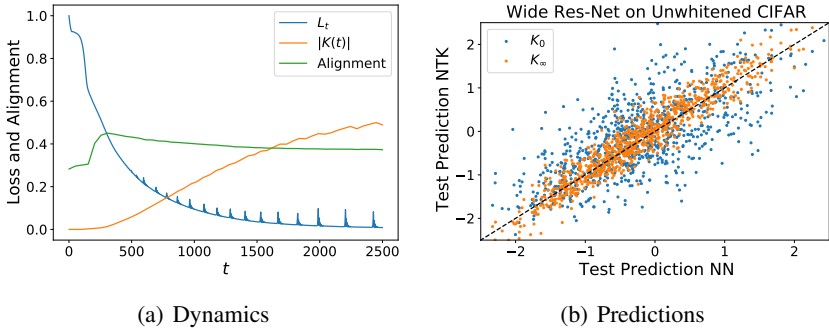

(a) Dynamics

(b) Predictions

Figure J.1: The dynamics and predictions of a Wide-Resnet with $k = 4$ and $b = 1$ on $P = 100$ unwhited CIFAR-10 images from the first two classes.

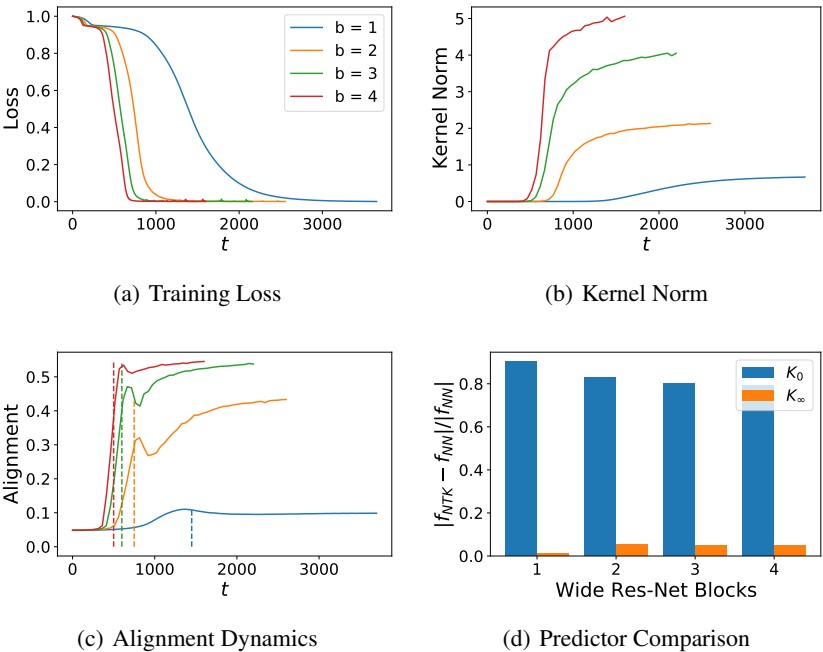

(a) Training Loss

(b) Kernel Norm

(c) Alignment Dynamics

(d) Predictor Comparison

Figure J.2: The silent alignment effect is preserved across a large range of depths in WideResNet trained on whitened CIFAR-10 images. The number of blocks per group $b$ alters the total number of conv layers ($6b + 1$ total conv layers). (a) The deeper models train faster with Adam. (b) The final NTK norm increases with depth. (c) The alignment achieves close to its final value by the time the kernel norm reaches $10\%$ of its final value (dashed line), indicating successful silent alignment. (d) The neural network predictions are very close to the predictions of the final NTK but is not accurately predicted by the initial NTK.

where $\eta_j(t)$ are time-varying functions which are computed in terms of the history of gradient moments for parameter $\theta_j$ or in terms of its instantaneous gradient. The relevant kernel at time $t$ which governs instantaneous evolution of network predictions is

$$K(\boldsymbol{x}, \boldsymbol{x}', t) = \sum_j \eta_j(t) \frac{\partial f(\boldsymbol{x}, t)}{\partial \theta_j} \frac{\partial f(\boldsymbol{x}', t)}{\partial \theta_j} \tag{91}$$

since $\dot{f}(\boldsymbol{x}) = \sum_\mu K(\boldsymbol{x}, \boldsymbol{x}^\mu, t)(y^\mu - f(\boldsymbol{x}^\mu, t))$. Though we do not calculate this kernel which is relevant to the adaptive learning rate scheme since it is not supported in Neural Tangents API, this could be a worthy future investigation.

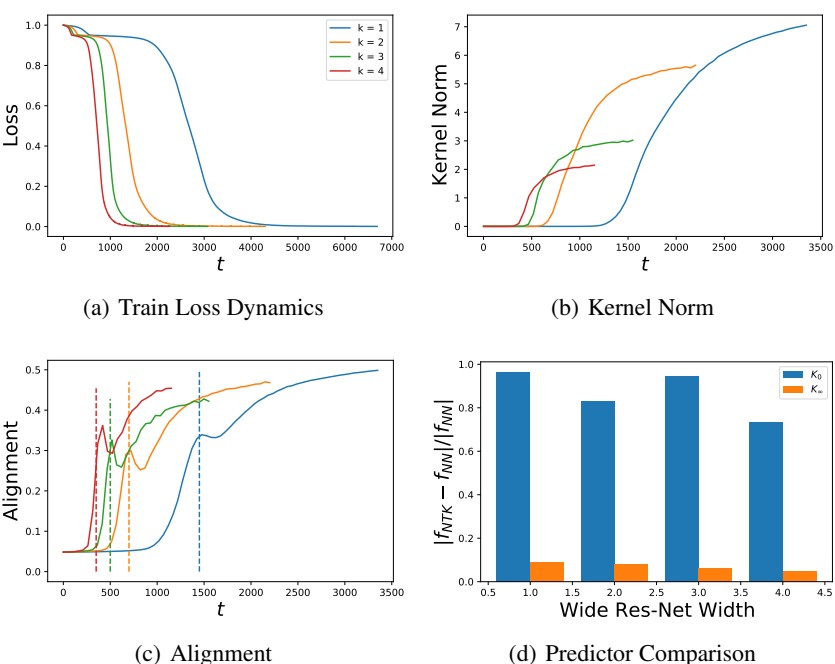

(a) Train Loss Dynamics

(b) Kernel Norm

(c) Alignment

(d) Predictor Comparison

Figure J.3: Varying the ResNet widening parameter $k$ also alters the kernel and loss dynamics. (a) The loss curve for $b = 2$ WRNs with widening factor $k$. Wider networks train more quickly. (b) The kernel norm increases more rapidly for wider networks but changes by a smaller amount. (c) Alignment reaches close to its asymptote by the time the kernel norm grows to $10\%$ its final value (dashed). (d) The final kernel is a much better predictor of the NN function than the initial kernel.

