# OpenReview forum: "Neural Networks as Kernel Learners: The Silent Alignment Effect"
_ICLR.cc/2022/Conference — ICLR 2022 Poster_

### Official Review · Reviewer_puin · 2021-10-26

**Correctness:** 4
**Technical Novelty And Significance:** 3
**Empirical Novelty And Significance:** 3
**Recommendation:** 5
**Confidence:** 4

**Main Review:**

Strengths:
-	The authors identify an interesting silent alignment effect, where the trained network acts as kernel predictor, and provide a thorough analysis for 2-layers linear networks and deep linear networks.
-	The authors demonstrate the effect empirically.

Weaknesses:
-	I think the authors should contrast their results with recent results on implicit bias of linear networks with small initialization. Specifically, it is shown in [1] (theorem 7 therein) and [2] (theorem 6.3 therein, for the case of 2 layers) that linear fully connected networks with small initialization converge to minimum $\ell_2$-norm solution (i.e. kernel solution with linear kernel). Note that this is correct also for anisotropic data.
-	The effect of depth is not so clear from the simulations. It can be useful to show on the same plot the kernel alignment for a fixed initialization scale but different depths.

Question:
-	Please explain why in Eq. (8) the error is $O(\sigma^2)$.
-	Please explain the transition in Eq. (15), why the integral starts from 0 and not from $\tau$ ?

Minor:
-	There is a typo in the definition of $K_{c,c'}$.
-	In figure 5, only one $\sigma$ is used.


[1] Yun et. al. A unifying view on implicit bias in training linear neural networks. ICLR 2021

[2] Azulay et. al. On the Implicit Bias of Initialization Shape: Beyond Infinitesimal Mirror Descent. ICML 2021


**Summary Of The Paper:**

In this paper the authors show that in some cases, a neural network, trained in the rich regime, can behave like a kernel predictor where the kernel is the neural tangent kernel at the end of training. Specifically, this happens when the kernel changes before the loss significantly decreases, but then stays fixed up to changes in scale. The authors call it “silent alignment”, and show that this happens for deep linear networks with small initialization and whitened data. Some experiments show that silent alignment can happen also for non-linear networks (e.g. with ReLU activations).

**Summary Of The Review:**

Overall the paper is interesting but the relation to implicit bias results is not discussed. I will be happy to adjust my score if the authors can address my questions, and also compare the results to known implicit bias results.

---

> ### Author Response · Authors · 2021-11-19
> **Response to Reviewer puin**
>
> We thank the reviewer for their thoughtful comments on our draft. We would like to take the opportunity to focus on the highlighted weaknesses, which we have taken care to address in the latest revision.
>
> 1. On the implicit bias of deep linear networks, our study agrees with the findings in [1,2] that all linear MLPs with small initialization find the pseudo-inverse solution regardless of whitening. In particular in Theorem 7 of [1], the learned network linear map $\beta = X(X^\top X)^{-1} y$ is the same as what is learned in the $\sigma \to 0$ limit of our theory. This can be seen easily from our balancing argument. We added Appendix G.1 explaining this result in detail.
> 	As a concrete example, consider the two layer network $f(x) = a^\top Wx$. Under the balancing condition, $W(t) = a(t) z(t)^\top$ for some unit vector $z(t)$. Note that by gradient flow, $\dot{W}(t) = a(t) \sum_{\mu} (y_\mu - f_\mu(t)) x_\mu^\top  = \dot{a}(t) z(t)^\top + a(t) \dot{z}(t)^\top$. Taking a left product with the unit vector $\hat{a}(t) = \frac{a(t)}{|a(t)|}$ pointing in the direction of $a(t)$, we find
> 	$$\dot z(t) = \sum_\mu (y_\mu - f_\mu(t)) x_\mu - \frac{\hat{a}(t)^\top \dot{a}(t)}{|a(t)|} z(t).$$
> If, $z(t) \in \text{span}\{x_1, ..., x_P \}$, then $\dot z(t) \in \text{span}\{x_1,...,x_P\}$ so $z(t)$ will remain in this subspace for all $t$. Further, from small initialization $\dot{W}(0) = a(0) \sum_\mu y^\mu x_\mu^\top + O(\sigma^3)$ showing that $z$ does in fact only initially evolve in the subspace of the data. Lastly, note that $\hat{\beta}(t) = |a(t)|^2 z(t)$ so $\beta$ points in the direction $z$. At convergence, the network must interpolate the training data, which implies $\hat{\beta} \cdot x_\mu = y_\mu$. The only vector in $\text{span}\{x_1,...,x_P\}$ which has this interpolation property is the pseudo-inverse solution.
> 	In linear networks, we do not argue that anisotropy prevents the final solution from being the minimum norm interpolator (pseudo-inverse solution). Rather, we showed that anisotropy can be at odds with the kernel growing only in scale late in training which is required for our proposed silent alignment effect. In a nonlinear network, failure for the kernel to remain aligned at late times can interfere with the final NN predictor being a kernel regression solution with $K_{\infty}$ as we show in Figure 4.
> 	In the latest submitted draft, we have taken care to highlight this point more carefully. We have added the above discussion (in the deep linear setting) to appendix G.1, and have added figure C.1 to demonstrate how in anisotropic data, the kernel can realign itself later in training to the minimum $\ell_2$ solution.
> 2. To improve our exploration on the effect of varying depth we added simulations of MLPs and Wide-resNets of varying depths in Appendices E and J. These show how loss and kernel dynamics change when depth is varied and additionally provides comparisons of the NTK predictors with the learned NN function.
>
> We will now like to answer the questions of the reviewer:
>
> 1. In equation 8, the reason why the error is $O(\sigma^2)$ is due to the fact that the initial kernel scales as $O(\sigma^2)$. Letting the kernel be written as $K(x,x') = x^\top M(t) x'$, we show that $M(t)$ evolves along a single direction $M'(t) \propto \beta\beta^\top + I$ early in training. However, the true kernel at time $t$ is $K =x' \int dt' M'(t) x + x' M_0 x$ which contains dependence on $M_0 = O(\sigma^2)$.
> 2. We thank the reviewer for the good catch on equation 15! The integral should start from $\tau$. We also had a typo on the definition of the $h$ function which should be $h(t) = \int_{\tau}^t g(s) ds$ so that $h'(t) = g(t)$ and $h(\tau) = 0$. This makes the change of variables in the last line of the derivation legitimate.
>
> We have further addressed the other minor typos highlighted by the reviewer, and thank them for the care and diligence they took in reading our draft.

---

> > ### Comment · Reviewer_puin · 2021-11-23
> > **Response**
> >
> > Thank you for the feedback.
> >
> > I read the updated draft and also the other reviews, and decided to keep my score as is due to the following reasons:
> >
> > I think the authors should be more precise about the contribution and results of the paper. For example, it is stated in the abstract: "We show that such an effect takes place in homogenous neural networks with small initialization and whitened data". However, the authors prove the silent alignment only for linear fully-connected networks. Also, it should be noted that for linear non-fully-connected networks (e.g. linear diagonal networks as in Woodworth et.al 2020) the final predictor may not be captured by a kernel.
> >
> > In addition, for depth>2 there is no clear proof that silent alignment occurs, since the authors show that theoretically there is no separation between the alignment phase and spectral learning phase.
> >
> > Finally, I think it would be good if the authors formalize the theoretical claims as theorems. I agree with reviewer XmeY that the analysis in the paper is not completely rigorous, even the updated draft. For example, it seems that some of the results are only valid for large dimension (appendix C.1) or large width (appendix H.2).

---

> > > ### Author Response · Authors · 2021-11-24
> > > **Response**
> > >
> > > 1. Your point about precision is well taken. It is true that our linear network analysis is restricted to MLPs: we do not consider tensor networks or diagonal networks etc. We made the following change to the abstract: "We *empirically* show that such an effect takes place in homogenous neural networks with small initialization and whitened data. We provide an analytical treatment of this effect in the *fully connected linear network* case." In this new sentence, we emphasize both that we provide only *empirical* evidence of early alignment for the nonlinear case (rather than proofs), and we only provide analytic treatment of the *fully connected linear case*. In the introduction, we modified the sentence "In Section 3, we provide an analysis of the NTK evolution of two layer *linear MLPs* with scalar target function with small initialization." and also provide the sentence "we extend our analysis to *deep linear MLPs* by showing that the time required for alignment scales with initialization the same way as the time for the loss to decrease appreciably. Still, these time scales can be sufficiently separated to lead to the silent alignment effect for which we provide empirical evidence."
> > > 2. While we do not provide analytic proof that alignment and loss timescales are separated in their $\sigma$ dependence in the deep $L>2$ linear MLP case, the timescales can still be separated through non-$\sigma$ dependent quantities: $t_{1/2} - t_{align} = \sigma^{-L+2} C$ for constant $C$. We do provide numerical experiments demonstrating that $C$ is positive for wide range of $L, \sigma$ (see Figure 3 b, and Figure E.1). Based on your concern, we have now studied this time gap for a range of network widths as well. A figure showing positive $C$ over a range of widths can be found [here](https://imgur.com/a/0yWNhAR) (it appears that very large width $N$ supresses separation time). We are currently working on a theory of phase I dynamics for the deeper linear MLPs, but it is much more involved than the two layer case since early time dynamics are nonlinear in the variables of interest.
> > > 3. Thank you for the advice, we will consider reorganizing the statement of our results into theorems rather than exposition. We mention that we focus on transparency and predictive power of our derived expression in numerical experiments over formality and rigor. Lastly we would like to stress that our results hinge neither on large width nor large input dimension. For section C.1, equation 22 is asympotically exact in the $\sigma,t\to0$ limit, provided that one has calculated $q_0, r_0$. We only consider large $D$ in equation 23 and 25 so that we can accurately characterize $q_0, r_0$ in terms of $N,D$ (as we successfully do to generate Figure C.3). We stress that this last approximation is not necessary to give an alignment time which is independent of initalization scale, nor to achieve a separation between alignment and spectral learning. Similarly, the extension in H.2 is not necessary to our analysis of kernel evolution since the error in the weight balancing is  $O(\sigma^2)$ regardless of width. The balancing approximation can always be controlled by making initialization small. In H.2, we merely wanted to capture leading order corrections to the learned weights/kernel.

---

> > > > ### Comment · Reviewer_puin · 2021-11-24
> > > > **Response**
> > > >
> > > > Thank you.
> > > >
> > > > I think formal statements and proofs can strengthen the paper.
> > > >
> > > > Another comment: In the proof in appendix B, it seems that $\tau$ is assumed to be fixed (independent of $\epsilon$). However, for depth>2, both $\tau$ and $\epsilon$ depend on the initialization scale $\sigma$, such that when $\sigma\rightarrow 0$, $\epsilon$ decreases but $\tau$ increases, so need to explain why the first integral in Eq. (20) is negligible.

---

> > > > > ### Author Response · Authors · 2021-11-24
> > > > > **response**
> > > > >
> > > > > Thank you very much for noticing this! This is a very good catch. In general, it is correct that $\tau$ should be thought of as dependent on $\sigma$. We will make this dependence explicit in Appendix B. For depth $L$ networks, we show in Figure 3 that the alignment timescale is $\tau \sim \sigma^{-L+2}$ whereas the size of the initial kernel is $\epsilon \sim \sigma^{2L-2}$. Taking a product of these gives $\tau \epsilon \sim \sigma^{L} \ll 1$, which can still be controlled by taking $\sigma \to 0$. We will update equation 18-20 so that this is handled properly.

---

> > > > > > ### Author Response · Authors · 2021-11-24
> > > > > > **response part 2: quick sketch for phase 1**
> > > > > >
> > > > > > We will also be more explicit about an argument for the phase 1 dynamics as well. Let $s$ be a positive number for which $K_0(t) \preceq s I$ for all $t\in(0,\tau)$ (the maximum eigenvalue of $K_0$ over the interval would work).  The training error $L_t = |\Delta|^2$ can be bounded by Gronwall's inequality
> > > > > > $\frac{d}{dt} |\Delta|^2 = - \epsilon \Delta K_0 \Delta \geq - \epsilon s |\Delta(t)|^2$ which implies that $L_{\tau} \geq \exp(-\epsilon s \tau  ) L_0$. By the above argument,  $\epsilon \tau \sim O(\sigma^L)$ so the training error cannot have changed by too much.

---

> > > > > ### Author Response · Authors · 2021-12-03
> > > > > **Updates to Appendix B**
> > > > >
> > > > > Though we are no longer able to provide updates to the current pdf of the paper, we did reorganize and make more clear the results of Appendix B and we would like to bring these changes to the attention of the reviewer.
> > > > >
> > > > > 1. As the reviewer suggested, we organized our results into lemmas and theorems as detailed below.
> > > > > 2. We made explicit the dependence of $\tau(\epsilon)$ on $\epsilon$
> > > > >
> > > > > The first equation in Appendix B is replaced with
> > > > > $K(x,x',t) = \begin{cases}
> > > > >       \epsilon K_0(x,x',t) & t \leq \tau(\epsilon) \\\\
> > > > >       g(t) K_{\infty}(x,x') & t > \tau(\epsilon)
> > > > >    \end{cases}$
> > > > >
> > > > > We then introduce Lemma 1 and Lemma 2, which collectively describe the dynamics of the network predictions on training points
> > > > >
> > > > > # Lemma 1
> > > > > Let $k_{0}$ represent the maximum operator norm of $K_0(t) \in \mathbb{R}^{P \times P}$ achieved on the interval $(0,\tau)$. Let $\Phi(t) \in \mathbb{R}^{P \times P}$ be the transition matrix for the linear dynamics so that $\frac{d}{dt} \Phi(t) = - \epsilon K_0(t) \Phi(t)$ and $\Phi(0) = I$. Then,
> > > > >     \begin{equation}
> > > > >         |\Phi(\tau) - \Phi(0)|_{op} < \epsilon \tau(\epsilon) k_0.
> > > > >     \end{equation}
> > > > >
> > > > > We prove this lemma using the triangle inequality $|\Phi(\tau) - \Phi(0)| = |\int_0^\tau \dot{\Phi}(t) dt | \leq \epsilon \int_0^\tau |K(t) \Phi(t)| dt$.  Since $|\Delta(\tau) - \Delta(0)| \leq |\Phi(\tau) - \Phi(0)| |\Delta(0)| \leq \epsilon \tau(\epsilon) k_0 |\Delta(0)|$, we have established that training predictions move very little for $t\in(0,\tau)$ as $\epsilon \to 0$ if $\tau(\epsilon) \to 0$.
> > > > >
> > > > > Next, we work out Lemma 2 which describes training predictions $\Delta(t)$ in the time window $t \in (\tau,\infty)$
> > > > > # Lemma 2
> > > > > Suppose that from $t\in(\tau,\infty)$ that $\Delta(t)$ obeys the dynamics $\frac{d}{dt} \Delta(t) = - h'(t) K_{\infty} \Delta(t)$ where $\Delta(\tau)$ is as in equation. Then, for all $t \in (\tau,\infty)$,
> > > > > \begin{aligned}
> > > > >     \Delta(t) =  \exp\left( - h(t)  K_{\infty}  \right) \Delta(\tau) =  \exp\left( - h(t)  K_{\infty}  \right) \left[  (y - f_0) +  O(\epsilon \tau(\epsilon) k_0) \right] .
> > > > > \end{aligned}
> > > > > This Lemma directly follows from integrating factor soltution to the ODE and the result of Lemma 1.
> > > > >
> > > > > Next, we use Lemma 2 to prove what the neural network predicts on an arbitrary test point $x$.
> > > > > # Theorem 1
> > > > > Let the kernel have dynamics
> > > > >
> > > > > $K(x,x',t) = \begin{cases}
> > > > >       \epsilon K_0(x,x',t) & t \leq \tau(\epsilon) \\\\
> > > > >       g(t) K_{\infty}(x,x') & t > \tau(\epsilon)
> > > > >    \end{cases}$
> > > > >  where $g(t)$ is a continuous, integrable function with $\lim_{t \to \infty} g(t) = 1$. The function learned by the neural network is
> > > > > \begin{align}
> > > > >     f(x) - f_0(x)  = k_{\infty}(x) \cdot K_{\infty}^{-1} y + O_{\epsilon}(\epsilon \tau(\epsilon) ).
> > > > > \end{align}
> > > > >
> > > > > This equation follows from integrating the dynamics $\frac{d}{dt} f(x) = k(x,t) \cdot \Delta(t)$ and applying the controlled approximations to $\Delta(t)$ obtained in Lemma 1 (for $t \in (0,\tau)$) and Lemma 2 (for $t \in (\tau,\infty)$).
> > > > >
> > > > > We prove this theorem by breaking the integral $\int_0^\infty k(x,t) \Delta(t) dt$ into two intervals: $(0,\tau)$ and $(\tau,\infty)$ and we invoke the facts obatined about $\Delta(t)$ from Lemmas 1 and 2.
> > > > >
> > > > > We hope that these changes will enhance clarity, readability and rigor. Thank you for the useful feedback on this point.

---

### Official Review · Reviewer_XmeY · 2021-11-02

**Correctness:** 2
**Technical Novelty And Significance:** 4
**Empirical Novelty And Significance:** 3
**Recommendation:** 6
**Confidence:** 4

**Main Review:**

Strengths:
1. Understanding the behavior of neural networks outside the NTK regime is a major question in the theory of deep learning. This paper gives a precise characterization of the kernel evolution in certain settings, in which the learning can be cleanly divided into two phases. The end result is that the final predictor is equivalent to a kernel regressor wrt a data-dependent kernel.
2. The paper also investigates how varying certain design choices (e.g. depth, relative learning rates, covariance of data) affects the silent alignment effect.

Weaknesses:
1. There appears to be a small mistake in the treatment of deep linear networks. Since different layers have different learning rates, the corresponding NTK should also be a weighted sum of the contributions from all layers, instead of an unweighted sum (Eq. (36)).
2. The analysis in the paper is not completely rigorous because of all the approximations used, and many times equations are used for approximate equalities. It would be much better if the approximations can be controlled explicitly and rigorously.
3. Figure 5b, 5c: why does silent alignment not clearly happen for deeper networks (i.e. alignment keeps improving while the loss is decreasing)? Does the theory predict this?
4. Overall speaking, the results are quite restricted since they are only for linear networks and whitened data. It would be very interesting if some characterizations can be given for the final NTK in non-linear networks.

Minor points:
1. page 5 line 5: "achieved low loss" --> "achieved high loss"?
2. Eq. (10): $K_0(x, x')$ missing? Approximation error missing?
3. Figure 5 caption: $10^{-2}, 10^{-1}, 10^{-2}$?

**Summary Of The Paper:**

This paper identifies a phenomenon called the "silent alignment effect," which happens in linear networks as well as ReLU networks on whitened data, with sufficiently small initialization of the network weights. In such a regime, the NTK of the network changes its eigenvectors to align with the problem structure at the beginning while remaining small in scale; then, the kernel essentially stops changing its eigenstructure and just grows in scale, eventually leading to a kernel regression solution wrt the NTK after the initial phase. The paper also demonstrates that non-whitened data can weaken this effect, for which the NTK needs to evolve later in training.

**Summary Of The Review:**

This paper makes some interesting and novel contributions towards understanding the learning dynamics of neural networks through the lens of NTK evolvement. On the other hand, the setting where a complete answer can be provided is restricted, and the analysis in the paper is not rigorous.

---

> ### Author Response · Authors · 2021-11-19
> **Response to Reviewer XmeY Part 1**
>
> We thank the reviewer for their thoughtful comments. We acknowledge that the limited amount of analytic control that we have over deep nonlinear networks is the primary reason for first identifying the silent alignment effect in the simpler linear case. In the more recent draft, we have taken care to put forward a larger class of experiments to demonstrate that the analytic predictions developed in the linear setting extend to more realistic ReLU MLPs (Appendix E), and that the silent alignment effect can even be observed for ResNET architectures trained on whitened data (Figure 1 and Appendix J for a broader range of experiments).
>
> We would like to take the opportunity to address the weaknesses raised in serial:
>
> 1. *There appears to be a small mistake in the treatment of deep linear networks. Since different layers have different learning rates, the corresponding NTK should also be a weighted sum of the contributions from all layers, instead of an unweighted sum (Eq. (36)).*
> 	We are especially grateful for the good catch that equation (36) should indeed include factors of the learning rates in the definition of the NTK. To avoid confusion, we have removed our discussion of the scaling with learning rate from the newer version. We have also added an appendix J.2 where we discuss the effect of optimizers such as Adam which employ adaptive per-parameter learning rates on the form of the neural tangent kernel which instantaneous evolution of predictions.
> 2. *The analysis in the paper is not completely rigorous because of all the approximations used, and many times equations are used for approximate equalities. It would be much better if the approximations can be controlled explicitly and rigorously.*
> 	We have added some more explicit discussion in the first paragraph of section 3 explicitly acknowledging that "though we have systematically tracked the error incurred at each step, we have focused on transparency over rigor in the following derivation." We have further revised the discussion of analytic solutions for linear networks in section 3 to enhance the rigor of our arguments and systematically tracked the scale of the error incurred at each step. Consequently, we have disposed of the use of the $\sim$ symbol. Our bounds on the error are sharp, since they rely on a conservation law on weights given in [1] that holds exactly, combined with an analysis of the early stage dynamics where higher order effects can be controlled. We have done the early stage analysis systematically in appendix C.2, providing an alternative derivation of the time range over which the phase one solution will provide accurate approximation to the dynamics.
> 3.  *Figure 5b, 5c: why does silent alignment not clearly happen for deeper networks (i.e. alignment keeps improving while the loss is decreasing)? Does the theory predict this?*
> 	The multiple timescales $s_\alpha$ for the multi-class deep linear network make it much harder to obtain a separation of timescales between alignment and spectral learning for depth greater than two. For a given singular value with index $\alpha$ both the the alignment time and the time to learn scale as $s_\alpha^{-1} \sigma^{2-L}$ as we show in appendix E.1. Consider the differences in alignment times $\Delta t_{\alpha \beta}^{(L)} = |t_{align,\alpha} - t_{align,\beta}|$ for two modes $\alpha,\beta$ with singular values $s_\alpha, s_\beta$. For a depth $L$ network, this difference in alignment times can be expressed as $\Delta t_{\alpha \beta}^{(L)} = \sigma^{-L+2} \Delta t_{\alpha \beta}^{(2)}$. The differences in alignment times between modes are thus exaggerated at larger depth for networks with small initialization. This exaggerated time difference can prevent the alignment times for different modes from clustering sharply before the first mode is learned like they do for the two-layer setting. Instead, we see that there are $C$ separate alignment stages, one for each singular value $s_{\alpha}$. This situation violates the silent alignment conditions. We have added this discussion in section H, acknowledging that silent alignment may fail for multi-class networks with L > 2. The figure has been moved to appendix H, where we describe the multi-class setting more thoroughly
> For scalar output deep networks, the effect persists much more prominently, and we have demonstrated this in figures E.1, E.2, E.3.
>
> [1] Du, Simon S., Wei Hu, and Jason D. Lee. "Algorithmic regularization in learning deep homogeneous models: Layers are automatically balanced." arXiv preprint arXiv:1806.00900 (2018).

---

> > ### Author Response · Authors · 2021-11-19
> > **Response to Reviewer XmeY Part 2**
> >
> > 4.  *Overall speaking, the results are quite restricted since they are only for linear networks and whitened data. It would be very interesting if some characterizations can be given for the final NTK in non-linear networks.*
> > 	We have added several appendices: E, J giving a much broader set of experiments. Therein, we test kernel alignment across ReLU MLPs, as well as a more realistic ResNET architecture. We also highlight the ResNET experiment in the revised Figure 1 of the paper. We have performed these and other new experiments on more diverse dataset sizes. In all cases, we show that the alignment grows before the kernel, and that consequently the final NTK matches the NNs prediction on test points to high accuracy, in contrast with the $t=0$ NTK.
> >
> > We have also taken care to address the minor points:
> > 1. The typo has been fixed.
> > 2. We thank the reviewer for pointing out the missing approximation error. We have added it in to equation 10.
> > 3. We have re-done this experiment now so that all variances have the property that $\sigma^L = 10^{-4}$. This is to have the function be of the same scale for these experiments.
> >
> > Once again we appreciate the diligence and thought that has gone into this review. We hope that the much larger set of experimental results across architecture and dataset, together with the refinement of the analysis for the linear case will address the objections raised.

---

> > > ### Comment · Reviewer_XmeY · 2021-11-30
> > > **Follow up**
> > >
> > > Thank you for the answers and the revisions. In light of the additional experimental results and discussions, I am increasing my score from 5 to 6.
> > >
> > > It appears that in the ResNet experiments (Figures J.1, J.2, J.3), the kernel alignment and growing phases are not separated, i.e. they happen at roughly the same time as the loss decreases. So it's not really "silent alignment" as the title suggests. That said, I do think that the finding in the linear network case is interesting and that the contributions in this paper could possibly be a good step toward understanding the NTK.

---

### Official Review · Reviewer_dMqU · 2021-11-02

**Correctness:** 4
**Technical Novelty And Significance:** 3
**Empirical Novelty And Significance:** 3
**Recommendation:** 8
**Confidence:** 3

**Main Review:**

## Summary

Kernel regimes have been a major workhorse in studying the training dynamics of deep neural networks. Previous work has shown that networks with large initialization scale and trained with gradient flow will converge to the kernel regression solution under a data-independent kernel, known as the NTK. However, it is now known that the practical training of neural networks often operates in a different regime called the "rich" regime, which makes it more difficult to analyze.

This paper bridges the gap between the two by pointing out that the rich regime can be split into two phases. The first phases is aligning the kernel spectrum with the training labels, which constructs a data-dependent kernel. The second phase only increases the scale of this kernel. This behavior is analytically shown on (deep) linear networks by writing out the dynamics of the NTK. Therefore, after the first phase, the neural network training will converge to the kernel regression solution of the adapted NTK. The same behavior, although not theoretically shown, are empirically verified on nonlinear ReLU nets.

## Strengths

* From the claims and the cited references this seems to be the first work that figures out the so-called "silent alignment" effect. The effect is analyzed on (deep) linear networks and is a bit surprising that how well it aligned with empirical evaluations.
* Combining the conservation law shown in Du et al. (2018) with early dynamics from small initialization seems to be the main "magic" that enables the analysis. The proof technique can itself be interesting to the community.
* I checked the derivation of major claims made in the paper and they looked correct to me.

## Weaknesses

* The analysis was done on (deep) linear networks. The justification on ReLU nets is through a combination of the observation of Chizat et al. (2019) and empirical evidence. It is unclear from the text if this assumption is widely tested on a variety of ReLU nets or just the single example of 1K whitened CIFAR images used in the paper. It would be nice to see if the silent alignment behavior is consistent across shallow/deep, narrow/wide nonlinear nets.

* The set-up of the simulation is a bit unclear to me. From the figures and captions I cannot see what the time span is for values like (t=1000). Does the results hold if the training spans as long as what people do in practice? The behavior might also change if the dataset contains way more patterns than the 1000 image experiments presented in the paper.

## Overall suggestions

I vote for acceptance of the paper. I should note that the submission should be best reviewed by deep learning theory people, since I am not particularly familiar with the latest literature that focuses on learning dynamics beyond the lazy regime. My assessment is based on the technical correctness and proper justification of the claims but may have missed some important related work that lowers the significance of contribution.

## Questions

* Figure 1b/1c: does the individual points correspond to training data or test data?
* Section 3.1: I don't understand what you mean by "In this setting the overall scale of the weights and kernel, evolves slowly since the network is initialized close to the saddle point at W, a=0".
* I don't see how the left hand side of (11) is approximated by the right.

## minor points

* Section 2: "However, we analytically show in sections 3 and 4 analytically"
* Missing period on page 5, last paragraph.

**Summary Of The Paper:**

Kernel regimes have been a major workhorse in studying the training dynamics of deep neural networks. Previous work has shown that networks with large initialization scale and trained with gradient flow will converge to the kernel regression solution under a data-independent kernel, known as the NTK. However, it is now known that the practical training of neural networks often operates in a different regime called the "rich" regime, which makes it more difficult to analyze.

This paper bridges the gap between the two by pointing out that the rich regime can be split into two phases. The first phases is aligning the kernel spectrum with the training labels, which constructs a data-dependent kernel. The second phase only increases the scale of this kernel. This behavior is analytically shown on (deep) linear networks by writing out the dynamics of the NTK. Therefore, after the first phase, the neural network training will converge to the kernel regression solution of the adapted NTK. The same behavior, although not theoretically shown, are empirically verified on nonlinear ReLU nets.

**Summary Of The Review:**

I vote for acceptance of the paper. It provides a new perspective of the "rich regime" of neural network training. The claims are backed by analysis on (deep) linear networks and align well with empirical evaluation.

---

> ### Author Response · Authors · 2021-11-19
> **Response to Reviewer dMqU**
>
> We thank the reviewer for their thoughtful comments and suggestions. Especially, we thank them for their feedback on some weaknesses of the paper that can be strengthened. In this latest submission we have addressed them as follows:
>
> 1. We have added new subfigures in Figure 1 of the main text and in Appendices E and J where we verify silent alignment across a much broader range of architectures and dataset sizes. Among these are varying depths of linear MLPs in E.1 and ReLU MLPs in E.2, as well as a Res-NET architecture in Figure 1 and various depths/widths of that in Appendix J. We have expanded the range of dataset sizes across these experiments to range from 100 to 3k datapoints. For example, the nonlinear MLP experiment in figure E.2 was trained with 3k datapoints from CIFAR-10. The bottleneck in scaling these experiments to even larger datasets is only the costliness of tracking the $P \times P$ kernel at every step.
> 2. Most experiments are done using full batch gradient descent, approximating gradient flow. The $t=1000$ means 1000 steps of gradient descent at the learning rate that we used. It is indeed the case that a larger dataset size requires a longer time to train if the learning rate is kept constant. We have also done new experiments for ReLU MLPs on larger datasets of 3k data points in Figure E.2 and observe qualitatively similar behavior.  Please let us know if we are misunderstanding this question.
>
> We will now take the opportunity to answer the three questions raised
>
> 1. *Figure 1b/1c: does the individual points correspond to training data or test data?*
> 	Indeed, figure 1b/1c plots the output of the trained network against both the initial and final NTKs on *test* data specifically. It is not a surprising fact that the final NTK and the trained network agree on train data, but the strong agreement on test data indicates that they are learning the same function.
> 2. *Section 3.1: I don't understand what you mean by "In this setting the overall scale of the weights and kernel, evolves slowly since the network is initialized close to the saddle point at W, a=0".*
> 	This sentence intended to convey that near the point of zero network connectivity, $\mathbf W, \mathbf a = 0$ (which is a saddle point of the loss function), the gradient of the network is small. Consequently, the weights evolve slowly under gradient flow. We have re-worded this sentence as follows: "In this setting, since the network is initialized near $\mathbf W, \mathbf a= \mathbf 0$, which is a saddle point of the loss function, the gradient of the loss is small. Consequently, the magnitudes of the weights and kernel evolve slowly."
> 3. *I don't see how the left hand side of (11) is approximated by the right.*
> 	We have added Appendix I to explicitly clarify the steps in the derivation of this approximation.
>
> We have further addressed the other minor typos highlighted by the reviewer, and thank them for the diligence they took in reading our draft.

---

> ### Comment · Reviewer_dMqU · 2021-12-02
> **Post rebuttal**
>
> I'd like to thank the authors for the additional experiments. It does show that the silent alignment effect is observed on ResNet in the whitened data setting. I'll keep my current score.

---

### Author Response · Authors · 2021-11-19
**Revisions to First Draft**

We thank all of the reviewers for their careful reading and useful suggestions. We have made a variety of updates to our paper in accordance with their reviews. The major changes are as follows:

1.  We included a much broader set of experiments, verifying silent alignment on MLPs and ResNET architectures. We performed this analysis across dataset sizes, depths, and widths on whitened data. The new MLP figures can be found in appendix E. There, we plot alignment curves for a fixed initialization scale at different depths in linear (Figures E.1, E.3) and ReLU (Figure E.2) networks. The new ResNet figures can be found in Appendix J, where we also study the alignment effect across of width and depth.
2. We added a comment about the inductive bias of linear networks with small initialization in section 3.2. In particular, in the Appendix G.1, we provide an alternative derivation of the fact that the linear networks from small initialization converge to the pseudo-inverse solution regardless of whitening. There is an additional visual in Figure C.1.
3. We tightened up Appendix C.2 to give more careful derivation of phase 1 dynamics. We also added studies of our analytic solutions vs empirical simulations in Figure C.3.
4. We clarified our discussion of several points in the appendix, including laziness, learning rate analysis, and generally replaced previous statements.
5. We discussed the extent to which we can control timescales of learning in the deep case, and provided emprical evidence for such a separation of timescales in the case of single-output networks.
6. We explain why multiple class learning problems in linear networks of depth greater than 2 will have separate alignment timescales for each singular values. This discussion can be found in Appendix H, which provides a simple explanation in the the difference between alignment dynamics in Figures H.1 a and H.1 b.
7. Finally, we have expanded the first section to include a much more thorough review of prior literature. The new Related Works subsection is more organized and comprehensive.

---

### Decision · Program_Chairs · 2022-01-20

**Decision:**

Accept (Poster)

**Comment:**

The authors make a case for a phenomenon of deep network training
that they call the "silent alignment effect": that, while the training
error is still large, the NTK associated with the network aligns its eigenvectors
with key directions in "feature space".  They support this with non-rigorous theoretical
analysis of linear networks, and extensive experiments with real networks on real data.
The consensus view was that this paper provides novel and useful insight into training
dynamics, in particular regarding feature learning.